# Spatially controlled construction of assembloids using bioprinting

Julien G. Roth [1,2,3], Lucia G. Brunel [4,9], Michelle S. Huang [4,9], Yueming Liu[5], Betty Cai[5], Sauradeep Sinha[6], Fan Yang[6,7], Sergiu P. Paşca [2,8], Sungchul Shin[5] & Sarah C. Heilshorn [2,5] ✉

The biofabrication of three-dimensional (3D) tissues that recapitulate organ-specific architecture and function would benefit from temporal and spatial control of cell-cell interactions. Bioprinting, while potentially capable of achieving such control, is poorly suited to organoids with conserved cytoarchitectures that are susceptible to plastic deformation. Here, we develop a platform, termed Spatially Patterned Organoid Transfer (SPOT), consisting of an iron-oxide nanoparticle laden hydrogel and magnetized 3D printer to enable the controlled lifting, transport, and deposition of organoids. We identify cellulose nanofibers as both an ideal biomaterial for encasing organoids with magnetic nanoparticles and a shear-thinning, self-healing support hydrogel for maintaining the spatial positioning of organoids to facilitate the generation of assembloids. We leverage SPOT to create precisely arranged assembloids composed of human pluripotent stem cell-derived neural organoids and patient-derived glioma organoids. In doing so, we demonstrate the potential for the SPOT platform to construct assembloids which recapitulate key developmental processes and disease etiologies.

The development of the human nervous system is predicated upon spatiotemporally controlled interactions between cells from distinct lineages[1]. These interactions occur early in gestation and are therefore inherently inaccessible for studies that probe neurodevelopmental phenomena or evaluate the efficacy of drugs targeting tissues in their native environment. Human neural organoids, three-dimensional (3D) stem cell-derived cultures that self-organize and exhibit tissue-mimetic cytoarchitecture and physiology, have been shown to recapitulate facets of brain development in vitro[2–5] and are beginning to reveal mechanistic insights into disease etiologies[6,7]. To model cell-cell interactions and circuit formation in the developing brain, multiple neural organoids have been fused into single integrated tissues known as neural assembloids[8–15]. Conventionally, neural organoid fusion is achieved by manually transferring organoids with a wide diameter pipette tip into a microcentrifuge tube containing culture medium where, over the course of several days, the constituent organoids integrate to form an assembloid[16]. While the construction of these structures enables temporal control of the interactions between organoids, multidimensional spatial control of their fusion remains a challenge.

The integration of distinct cell types into organoids is broadly relevant outside of recapitulating neurodevelopmental phenomena and probing neuropsychiatric disease etiology. For example, organoid-based cancer models have emerged as a promising platform for maintaining inter- and intratumoral heterogeneity, enabling ex vivo investigation of patient-specific tumor progression[17,18]. To date, two

[1]Institute for Stem Cell Biology and Regenerative Medicine, Stanford University School of Medicine, Stanford, CA, USA. [2]Stanford Brain Organogenesis, Wu Tsai Neurosciences Institute & Bio-X, Stanford University, Stanford, CA, USA. [3]Complex in Vitro Systems, Safety Assessment, Genentech Inc., South San Francisco, CA, USA. [4]Department of Chemical Engineering, Stanford University, Stanford, CA, USA. [5]Department of Materials Science and Engineering, Stanford University, Stanford, CA, USA. [6]Department of Bioengineering, Stanford University, Stanford, CA, USA. [7]Department of Orthopedic Surgery, Stanford University School of Medicine, Stanford, CA, USA. [8]Department of Psychiatry and Behavioral Sciences, Stanford University, Stanford, CA, USA. [9]These authors contributed equally: Lucia G. Brunel, Michelle S. Huang. ✉e-mail: heilshorn@stanford.edu

main approaches have been developed for recapitulating the tumor-host cellular microenvironment in vitro: (i) by leveraging genetic engineering strategies to induce oncogenic mutations, and (ii) by co-culturing tumor cells with organoid models of the tissue of origin or the tissue of metastasis. These models permit temporal control over the interactions between tumor and host tissue, yet they offer limited spatial control of the juxtacrine and paracrine signaling within the tumor microenvironment.

3D bioprinting, a process wherein cells, often with accompanying biomaterials, are deposited and assembled into tissues, has been leveraged to control the spatial arrangement of spheroids and organoids. Early descriptions of spheroid bioprinting demonstrated the layer-by-layer extrusion of cellular aggregates or cylindrical rods[19–23]. While pioneering, these approaches employed primary cell spheroids that were devoid of internal cytoarchitecture, generally limited in diameter to under 500 μm, and expected to exhibit standardized sizes such that nozzle clogging was obviated[24]. Since then, the printing of organ building blocks (OBBs) has been broadly categorized into two distinct approaches[25]: continuous bioprinting, wherein the OBBs are encapsulated within the bioink or support scaffold[26,27], and aspiration-assisted bioprinting (AAB), wherein individual OBBs are manipulated by vacuum pressure[28,29]. Continuous bioprinting of neural organoids, while capable of creating thick, patterned tissue structures[27], is limited by its inability to address the positioning of individual OBBs as well as the high cost associated with deriving enough OBBs to populate the bioink or scaffold. While significantly lower-throughput, AAB would be better suited to spatially pattern the fusion of neural assembloids in 3D as it is capable of controlling the specific 3D position of each OBB. However, here, we demonstrate that AAB is poorly suited for the fabrication of neural assembloids, as neural organoids exhibit large diameters, relatively weak surface tension, and a propensity to undergo plastic deformation and degrade under relatively low vacuum force.

In this work, we develop an approach we term Spatially Patterned Organoid Transfer (SPOT) to facilitate the construction of neural assembloids in 3D with fine spatial control over OBB fusion. SPOT employs a magnetic nanoparticle (MNP)-laden cellulose nanofiber (CNF) hydrogel, a CNF support scaffold enclosed within a custom-designed reservoir, and a magnetized 3D printer to control the spatial arrangement of the OBBs. Once fused, the resultant assembloid can be released from the support with bioorthogonal, on-demand degradation of the CNF scaffold. We leverage SPOT to control the spatial position of OBBs in two classes of neural assembloids. Firstly, for assembloids employed in studies of neurodevelopmental phenomena, we leverage SPOT to facilitate the construction of assembloids composed of dorsal and ventral forebrain organoids, which mediate in vitro studies of the migration and integration of interneurons into the cortex. Secondly, for assembloids employed in translational studies of disease progression and drug efficacy, we leverage SPOT to create tissues in which human brain tumor organoids are integrated into neural organoids. Taken together, we demonstrate the potential for SPOT to precisely and reproducibly control the spatial dynamics of assembloid construction and, as such, to serve as a powerful platform for building complex in vitro models of the human brain.

## Results

### Physical characterization of hiPSC-derived neural organoids

Previous demonstrations of AAB reported a linear relationship between the diameter of OBBs and their required lifting pressure[28]. To date, most demonstrations of AAB include mesenchymal stromal cell (MSC) spheroids as the OBBs[28–31] (Supplementary Table 1). As such, we began our characterization of the physical properties of OBBs by comparing human forebrain neural organoids to MSC spheroids. We utilized established differentiation methods to generate regionalized human induced pluripotent stem cell (hiPSC)-derived dorsal (cortical) and ventral (subpallium) forebrain organoids[8]. These organoids

exhibit canonical markers of dorsal progenitor and ventral forebrain cell fate (Supplementary Fig. 1). Previous studies have observed that the optimal time window during which such organoids should be fused is between days 50 and 90 of differentiation[16,32]. Compared to MSC spheroids ($270.5 \pm 15.2$ μm, mean ± SD) and human umbilical vein endothelial cell (HUVEC) spheroids ($240.8 \pm 21.8$ μm), hiPSC-derived forebrain neural organoids are already significantly larger in diameter at day 25 (ventral: $1143 \pm 121.2$ μm, dorsal: $1551 \pm 179.9$ μm, $p < 0.0001$). While MSC and HUVEC spheroids showed negligible change in diameter over time, neural organoids continued to increase in size both at day 50 (ventral: $1492 \pm 161.4$ μm, dorsal: $2635 \pm 216.1$ μm) and day 100 (ventral: $1770 \pm 226.7$ μm, dorsal: $2830 \pm 182.2$ μm; Fig. 1a). As the diameter of these neural organoids increased, so too did their mass (Fig. 1b). Consistent with previous reports[28], these increases in size led to concomitant increases in the minimum pressure required to lift the submerged tissue using AAB from $1.4 \pm 0.3$ mmHg to $6.3 \pm 0.3$ mmHg (Fig. 1c).

The structural integrity of OBBs lifted with AAB is dependent upon the degree to which the tissue is resistant to deformation by aspiration associated forces. To characterize this resistance, we used micropipette aspiration, a technique originally developed to measure the surface tension of single cells[33,34]. We applied a similar micropipette aspiration protocol to our organoids to calculate apparent surface tension, as it is a reproducible measurement and has previously been reported for spheroids[28,30] (Supplementary Fig. 2a, b). Although neural organoids require higher lifting pressures than MSC spheroids, they have significantly lower apparent surface tension ($p < 0.0001$; Fig. 1d). Taken together, these observations imply that neural organoids may experience marked structural deformation when manipulated by AAB (Fig. 1e).

### Neural organoid deformation following AAB and SPOT

To evaluate potential deformation during aspiration, we exposed neural organoids to their minimum lifting pressure. Following aspiration, the surface of the neural organoids exhibited substantial local distension (Fig. 1f–h). This deformation increased as a function of the applied pressure across multiple replicates (Fig. 1i). Once the aspiration force was released, neural organoids continued to briefly distend before retracting; interestingly, even at the minimum lifting pressure, this retraction was incomplete, and irreversible plastic deformation was observed (Fig. 1j). While neural organoid viability does not seem to vary (Supplementary Fig. 3), these plastic deformations were substantial at both 6 mmHg ($491.9 \pm 178.4$ μm) and 10 mmHg ($652.2 \pm 232.3$ μm; Fig. 1k). After 15 days, 75% of organoids still exhibited protrusions greater than 50 μm (Supplementary Fig. 4a, b). Moreover, 20% of organoids underwent severe distension that distorted their spherical shape following release of the aspiration force (Supplementary Fig. 4a. c). Importantly, these macroscopic deformations were associated with striking microscopic changes in cellular organization, namely the disintegration of the canonical ventricular zone (VZ)-like structures with PAX6-expressing progenitors radially arrayed around lumens lined by NCAD-expressing cells (Supplementary Fig. 5). Over time, intact VZ-like structures give rise to concentric rings of deep subcortical projection neurons and superficial cortical neurons in a manner that resembles the cortical layers of the developing human brain[2]. Given the imperative of a conserved cytoarchitecture within the neural organoid[35,36], this degree of distension would be prohibitive to studies of neural development or disease.

The SPOT platform relies upon the magnetic actuation of MNPs, which are embedded within a bioinert CNF ink biomaterial that encases the OBB. An iron rod affixed to an electromagnet mounted on a modified 3D printer is used to control the lifting, positioning, and deposition of the MNP-coated OBB. As such, with SPOT, the OBB is lifted in response to a force that is distributed across the entire OBB surface, unlike aspiration, which concentrates force and results in

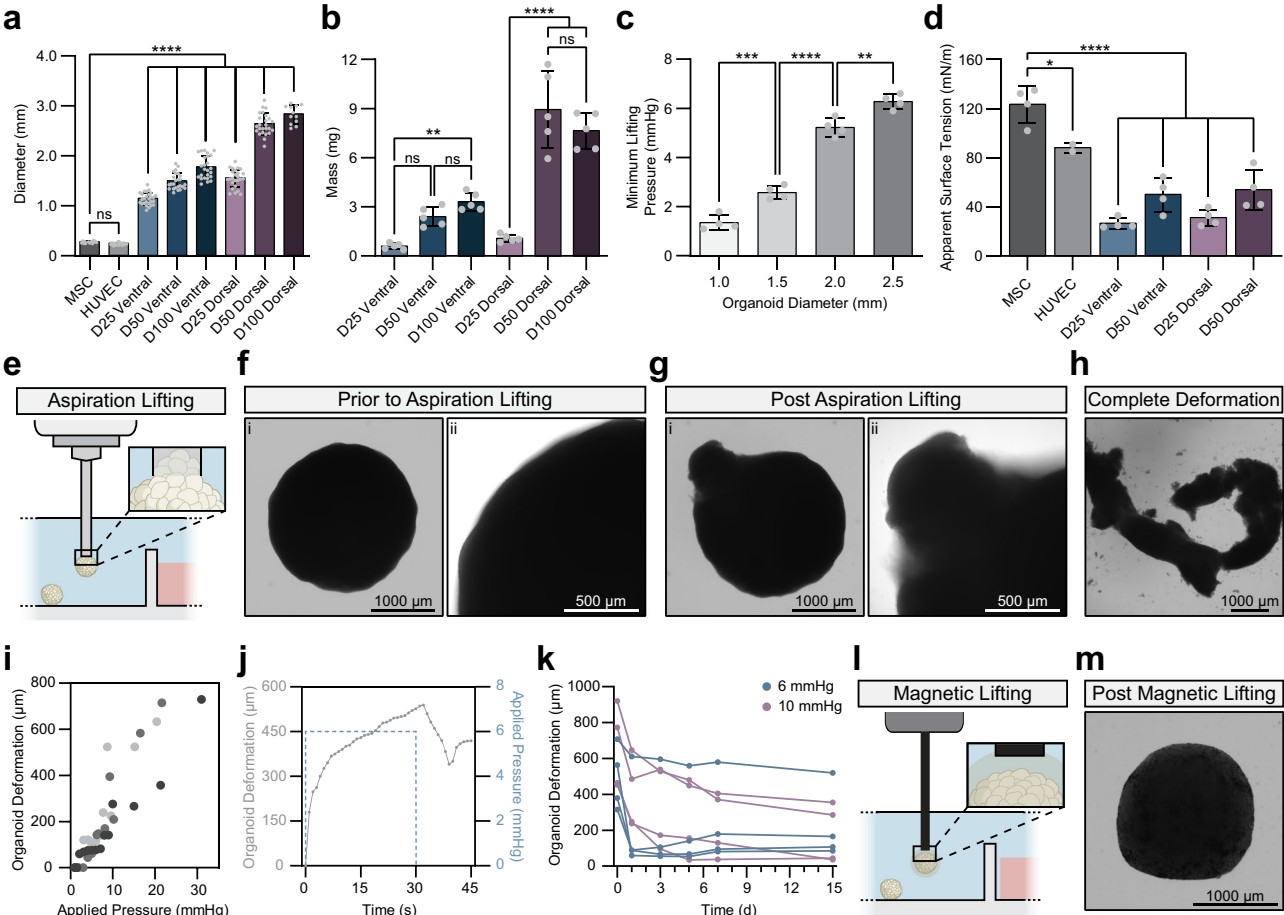

**Fig. 1 | Magnetic lifting maintains the structural integrity of neural organoids.**
**a** Diameter measurements of MSC and HUVEC spheroids, and hiPSC-derived ventral and dorsal forebrain neural organoids at increasing days of culture. Each data point represents a distinct spheroid or organoid (MSC $n = 25$, HUVEC $n = 27$, D25 Ventral $n = 27$, D50 Ventral $n = 25$, D100 Ventral $n = 25$, D25 Dorsal $n = 25$, D50 Dorsal $n = 25$, D100 Dorsal $n = 12$). $p$ values for each diameter comparison are as follows: MSC vs. HUVEC $p = 0.9978$, all other shown comparisons $p < 0.0001$. **b** Mass measurements of spheroids and neural organoids. Each data point represents an average of five neural organoids. $p$ values for each mass comparison are as follows: D25 Ventral vs. D50 Ventral $p = 0.1499$, D25 Ventral vs. D100 Ventral $p = 0.0094$, D50 Ventral vs. D100 Ventral $p = 0.7933$, D25 Dorsal vs. D50 Dorsal $p < 0.0001$, D25 Dorsal vs. D100 Dorsal $p < 0.0001$, D50 Dorsal vs. D100 Dorsal $p = 0.4743$. **c** Vacuum pressure required to lift neural organoids of increasing diameters within a liquid medium. Each data point represents a distinct organoid (1.0 mm $n = 4$, 1.5 mm $n = 4$, 2.0 mm $n = 4$, 2.5 mm $n = 4$). $p$ values for minimum lifting pressure comparisons are as follows: 1.0 mm vs. 1.5 mm $p = 0.0007$, 1.5 mm vs. 2.0 mm $p < 0.0001$, 2.0 mm vs. 2.5 mm $p = 0.0026$. **d** Apparent surface tension of spheroids and neural organoids. Each data point represents a distinct spheroid or organoid (MSC $n = 4$, HUVEC $n = 2$, D25 Ventral $n = 4$, D50 Ventral $n = 4$, D25 Dorsal $n = 4$, D50 Dorsal $n = 4$). $p$ values for

apparent surface tension comparisons are as follows: MSC vs. HUVEC $p = 0.0235$, all other shown comparisons $p < 0.0001$. **e** Schematic of vacuum aspiration-assisted lifting of neural organoids. **f** Representative brightfield (BF) images of a neural organoid prior to vacuum aspiration. **g** Representative BF images of a neural organoid post vacuum aspiration (6 mmHg). **h** Representative BF image of a neural organoid that has undergone complete deformation (i.e., is no longer spherical) post vacuum aspiration (6 mmHg). **i** Quantification of the extent of deformation as a function of the applied vacuum pressure. Each color represents a single neural organoid ($n = 3$). **j** Representative quantification of neural organoid deformation during and immediately following vacuum aspiration (6 mmHg, shown in blue). **k** Long-term neural organoid deformation in response to two vacuum pressures: 6 mmHg and 10 mmHg. Each set of data points connected with a line represents a single biological replicate (6 mmHg $n = 4$, 10 mmHg $n = 4$). **l** Schematic of magnetic lifting of neural organoids. **m** Representative BF image of a neural organoid post magnetic lifting. Statistical analyses performed as one-way ANOVA with Tukey multiple comparisons test. Unless otherwise noted, all data points represent distinct biological replicates. Data plotted as mean ± SD where *$p < 0.05$, **$p < 0.01$, ***$p < 0.001$, ****$p < 0.0001$, and ns not significant.

deformation during the lifting process (Fig. 1e, l). As a result, structural deformation with SPOT is not observed (Fig. 1m).

## SPOT facilitates the controlled lifting, transfer, and deposition of neural organoids in 3D

The SPOT platform consists of the following series of repeatable, automatable steps: (i) coat organoids with the iron-oxide MNP embedded CNF ink, (ii) lift the coated organoid with an iron rod attached to an electromagnet-modified 3D printer, (iii) position the lifted organoid in 3D within a CNF support scaffold, and (iv) turn off the electromagnet and remove the iron rod (Fig. 2a, b).

The organoid coating process is achieved by first mixing MNPs into a CNF hydrogel and then dispensing fixed volumes of the mixture atop each individual OBB. Both commercially available MNPs and iron-oxide nanoparticles synthesized in-house through co-precipitation in a basic solution (Supplementary Table 2) were successfully utilized for OBB coating. After 30 minutes of coating, the MNPs are evenly distributed across the surface of the organoid (Supplementary Fig. 6a). To lift the coated organoid, a conventional 3D printer is modified such that an affixed electromagnet can be switched on and off with the same G-code that is used to direct the movement of the print head. An iron rod (here, with a diameter of 2 mm) is then bound to the

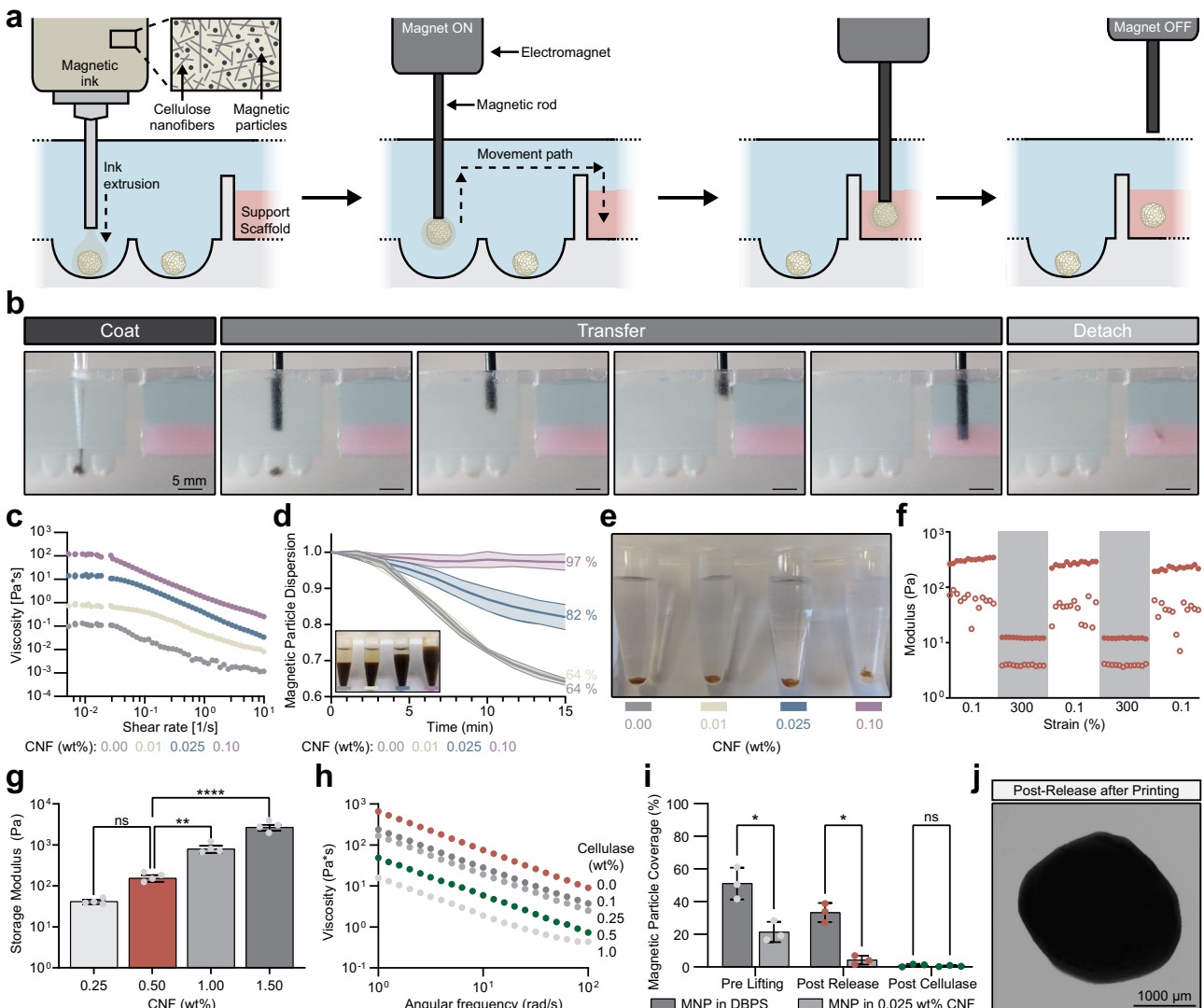

**Fig. 2 | Cellulose nanofibers mediate the bioprinting of neural organoids with SPOT. a** Schematic of the SPOT platform. **b** Representative images of a neural organoid being coated in a magnetic nanoparticle (MNP)-laden CNF ink, lifted and transferred into a CNF support scaffold by a magnetic rod attached to an electromagnet-modified 3D printer, and released at a desired position within the support scaffold. **c** Representative viscosity measurements of inks with 1 wt% MNP and various CNF wt%. **d** Quantification of the relative degree of MNP dispersion within CNF inks of various wt% over 15 minutes (n = 4 formulations); error bars represent standard deviation. Inset: Representative image of MNP dispersion within a microcentrifuge tube. **e** Representative image of MNP-laden CNF inks extruded over the top of neural organoids. **f** Representative storage modulus (filled circles) and loss modulus (open circles) of 0.5 wt% CNF support scaffold exposed to cyclical periods of low (0.1%) and high (300%) strain to evaluate the ability of the material to shear thin and self-heal. Percent G′ recovery following one cycle (mean ± SD): 86.8 ± 6.0 (n = 4 gels); percent G′ recovery following two cycles (mean ± SD): 84.3 ± 25.6 (n = 4 gels). **g** Storage modulus of CNF support scaffolds of various wt%.

Each data point represents a distinct gel (n = 4 for all formulations). p values for storage moduli comparisons are as follows: 0.25 wt% CNF vs. 0.50 wt% CNF p = 0.9001, 0.50 wt% CNF vs. 1.00 wt% CNF p = 0.0097, 1.00 wt% CNF vs. 1.50 wt% CNF p < 0.0001, **h** Representative viscosity measurements of 0.5 wt% CNF support scaffolds in response to treatment with various concentrations of cellulase. **i** Quantification of the extent of MNP coverage on the surface of a neural organoid following coating with 1 wt% MNPs in DPBS or a 1 wt% MNP-laden 0.025 wt% CNF ink. Each data point represents a different organoid (n = 3). p values for MNP coverage comparisons are as follows: Pre-Lifting p = 0.048, Post-Release p = 0.0162, Post-Cellulase p = 0.851. **j** Representative BF image of a neural organoid following SPOT. Statistical analyses were performed as one-way ANOVA with Tukey multiple comparisons test or two-way ANOVA with either Dunnett's multiple comparisons test or Šídák's multiple comparisons test. Unless otherwise noted, all data points represent distinct biological replicates. Data plotted as mean ± SD where *p < 0.05, **p < 0.01, ***p < 0.001, ****p < 0.0001, and ns not significant.

electromagnet. Once the rod is bound, it is positioned above, and subsequently lowered toward, the coated organoid. As a function of the magnetic field strength, which is tuned by modulating the voltage of the electromagnet and the distance of the magnetic rod (Supplementary Fig. 7), the MNPs within the CNF ink are pulled towards the rod, resulting in lifting of the OBB (Supplementary Fig. 6b). As an alternative OBB coating approach that would be amenable to cultures grown within bioreactors, MNPs can be added directly to the medium of a suspension culture and agitated with an orbital shaker

(Supplementary Fig. 8). Once the organoid is affixed to the end of the magnetized rod, it can be transported from the liquid medium into the CNF support scaffold. While the CNF support scaffold is directly adjacent to the organoids in the setup shown here (Figs. 2a, b), any configuration in which the organoids can be transferred while remaining submerged within cell culture medium is amenable to SPOT. Importantly, the final position of the organoid can be addressed in X, Y, and Z dimensions. Once the desired position is achieved, the electromagnet is turned off, and the iron rod is removed.

## Characterization of the CNF ink and embedded MNPs

To support magnetic bioprinting, a potential cytocompatible ink material should: (i) undergo viscous thinning under an applied shear to allow for continuous dispensing of an MNP-laden ink through a syringe (ii) have a zero-shear viscosity that prevents MNP sedimentation over time scales relevant to coating multiple OBBs, (iii) encase the organoid fully once dispensed, and (iv) limit the degree of direct MNP contact with the organoid surface. A 0.025 percent by weight (wt%) solution of CNF exhibited a shear-thinning viscosity and significantly reduced MNP settling compared to 0.00 and 0.01 wt% CNF solutions (0.00 wt%: $64.1 \pm 0.9\%$ dispersed, 0.01 wt%: $64.4 \pm 0.6\%$ dispersed, 0.025 wt%: $82.0 \pm 3.5\%$ dispersed over 15 minutes, $p < 0.0001$; Figs. 2c, d). While the 0.10 wt% CNF ink solution resulted in significantly less MNP settling ($97.3 \pm 2.3\%$ dispersed, $p < 0.0001$), it did not adequately encase the organoid due to its higher viscosity and, therefore, was less well suited to reproducible organoid lifting (Fig. 2e). Following 30 minutes of incubation, the 0.025 wt% CNF ink uniformly coated the organoid with MNPs (Supplementary Figs. 6a and 9a). Qualitative evaluation of potential MNP uptake using Prussian blue staining of OBB cross-sectional slices demonstrated that the MNPs were primarily located at the periphery of the organoid without extensive intracellular localization (Supplementary Fig. 9b). While iron was detected throughout the coated organoid, a similar distribution was observed in control neural organoids that had never been exposed to MNPs (Supplementary Fig. 9c). Although iron oxide nanoparticles were previously shown to affect the MAPK signaling pathway in bone-derived MSCs[37], MAPK signaling in hiPSC-derived dorsal forebrain organoids was not affected by MNP surface coverage with SPOT (Supplementary Fig. 9d).

## Characterization of the CNF support scaffold and organoid release

In biofabrication, support scaffolds temporarily maintain the spatial positioning of cells within 3D space. To achieve this, support scaffolds must: (i) be shear-thinning so that the material yields as a deposition tool moves through the scaffold, and (ii) be self-healing after the deposition tool has passed to provide physical confinement to the OBB[29,38]. To create an optimal support scaffold, we sought to identify a material that was cytocompatible, bioinert to mammalian cells, and amenable to on-demand solubilization to release the encapsulated cellular structure after fabrication.

We evaluated the viscoelastic properties of a range of CNF solutions for use as a support scaffold. A concentration of 0.50 wt% CNF exhibited shear-thinning and self-healing properties without the need for additional chemical modifications or formulation additives (Fig. 2f). Moreover, the 0.50 wt% CNF demonstrated a greater recovery of modulus after high strain compared to 0.25 wt%, 1.0 wt%, or 1.5 wt% CNF (Fig. 2f and Supplementary Fig. 10). The reported stiffness of neural tissue varies as a function of sample age, brain region, and testing method, yet most studies report shear moduli ranging from several hundred to a few thousand pascal (Pa)[39–41]. The wt% of CNF can be tuned to reproducibly vary the stiffness of support, and the 0.5 wt% CNF had a plateau storage modulus (G') of approximately 150 Pa (Fig. 2g).

After being positioned, the fusion of constituent OBBs into a single assembloid can require multiple days during which the support scaffold should remain intact. The 0.5 wt% CNF support scaffold displayed a consistent range of storage moduli over 72 h, both with and without daily media changes (Supplementary Fig. 11a). Furthermore, the 0.5 wt% CNF support scaffold allowed for consistent media diffusion (Supplementary Table 3). After OBB fusion, the resultant assembloid can be removed from the scaffold for downstream applications. Although recent efforts have begun to introduce polymers into the medium of regionalized neural organoids[42], to date, most studies have cultured organoids in suspension without the addition of exogenous biomaterials[36]. CNF, as a cellulose derivative, is amenable to cellulase-

mediated degradation (Supplementary Fig. 11b). Treating the CNF support with a range of cellulase concentrations resulted in stepwise decreases in both viscosity and storage modulus over time (Fig. 2h and Supplementary Fig. 11c, d). Importantly, as cellulase activity is bioorthogonal to mammalian cultures, the addition of cellulase does not affect organoid viability (Supplementary Fig. 11e). Once released from the CNF support bath with 0.5 wt% cellulase, neural organoids may remain sparsely coated in residual CNF; however, the addition of 0.5 wt% cellulase removes over 98% of the material over 72 hours (Supplementary Fig. 12). In lieu of cellulase-mediated degradation, organoids can be released from their support scaffold through the gentle dilution of the CNF hydrogel with DPBS. Alternatively, organoids may be cultured within the scaffold for protracted periods of time.

Following the entire coating, lifting, transportation, deposition, and removal process, the majority of MNPs located on the surface of the neural organoids were no longer present (Supplementary Fig. 13). Moreover, by initially coating an organoid with an MNP-laden CNF bioink, as opposed to MNPs in solution, the degree of MNP attachment to the organoid surface significantly decreased (prior to lifting: $p < 0.05$ between MNP and MNP in CNF, post release from CNF: $p < 0.05$ between MNP and MNP in CNF, post cellulase treatment: $p = 0.851$) (Fig. 2i). Finally, following SPOT, the neural organoids appear devoid of the gross deformations observed with AAB (Fig. 2j).

## Utilizing SPOT to construct dorsal-ventral forebrain assembloids

The construction of multi-region neural assembloids that begin to recapitulate the circuitry of the developing brain was first demonstrated in a collection of studies in 2017[8–10]. Since then, increasingly complex assembloids have revealed compelling, heretofore unobserved, disease-relevant phenotypes in vitro[11]. To demonstrate the unique capabilities of the SPOT platform, we sought to create dorsal-ventral forebrain assembloids with precise, reproducible control over the 3D positioning of the constituent OBBs.

Each stage of OBB lifting, transportation, and deposition with SPOT can be performed manually or through automation. To facilitate automation, we created a custom PDMS chip with uniformly spaced wells and a support scaffold reservoir (Supplementary Fig. 14a, b). The chip includes several design elements, namely an offset platform for repeated medium addition, a series of elongated U-bottom wells, and a raised connector channel; collectively, these elements facilitate organoid maintenance prior to fusion, allow homogeneous MNP ink distribution during the coating phase, and maintain the OBB within a fully submerged medium, respectively. As the chip itself is fabricated from a 3D-printed mold, it can be scaled in size to accommodate a wide assortment of assembloid sizes and shapes. The automation of the SPOT assembly process can be controlled by G-code, a widely used computer numerical control programming language, which reproducibly locates the organoids in the chip, lifts and deposits them within the support bath, and releases them at a user-specified location. G-code scripts are passed from a laptop (or microSD card) to a 3D printer that has been modified such that the conventional fan controls now trigger a solid-state relay to activate the electromagnet (Supplementary Fig. 14c). Here, to support the potential for SPOT to be automated, we demonstrate G-code mediated control of (i) the extrusion of the magnetic ink over individual microwells, (ii) the movement of the magnetized rod between said microwells and the reservoir, and (iii) the simultaneous switching of the electromagnetic field on and off (Supplementary Movies 1 and 2). We also provide the accompanying G-code scripts (Supplementary Methods 1 and 2). When taken together, these components of the SPOT platform facilitate the reproducible, automatable construction of assembloids (Fig. 3a).

Spatial control over the position of constitutive OBBs is contingent on the precision, in all three dimensions, of both the initial placement

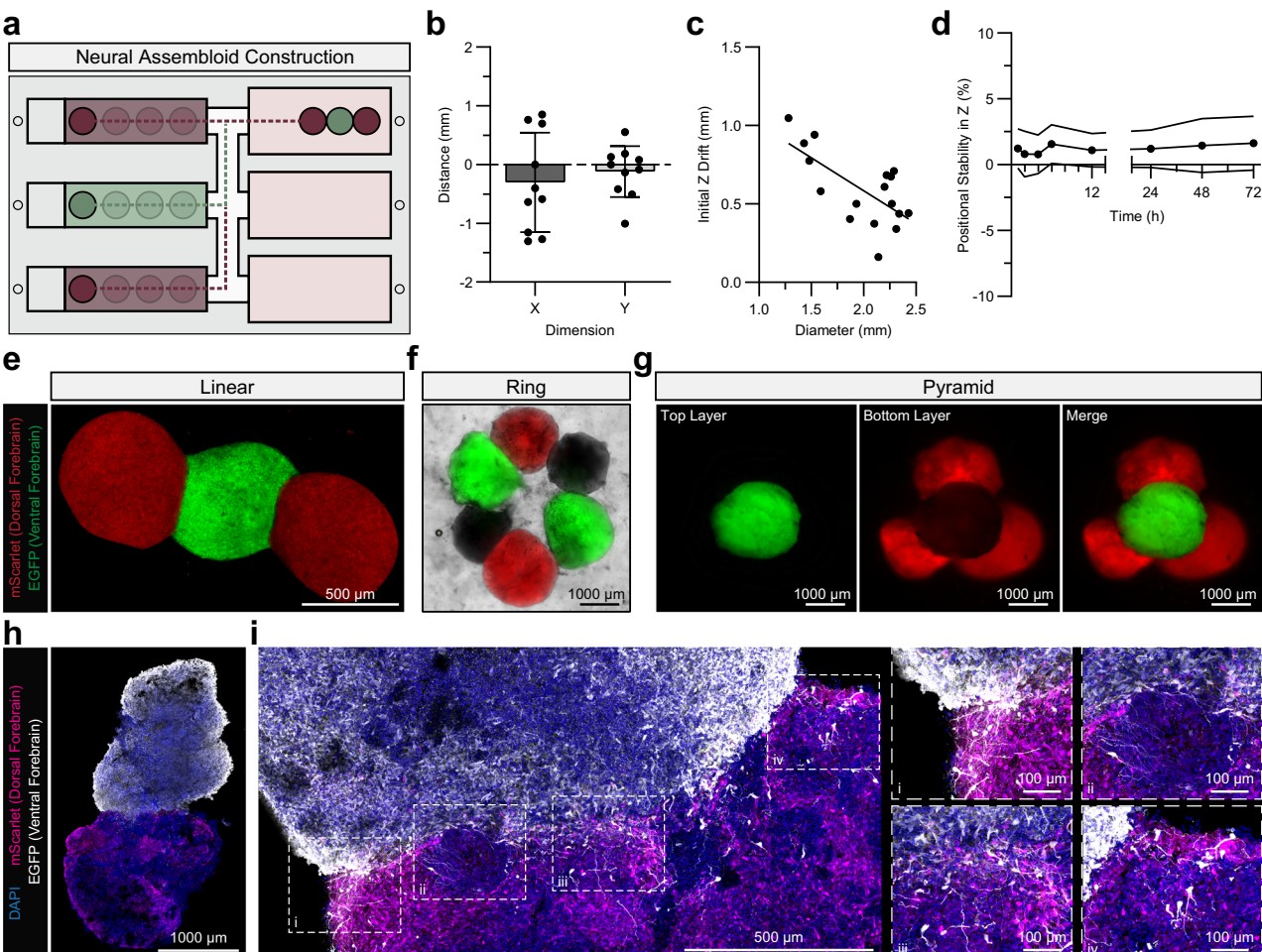

**Fig. 3 | SPOT imparts spatial control over the construction of neural assembloids. a** Schematic of the potential automation of SPOT. Specific media formulations (depicted as red or green) can be portioned into distinct channels within a custom-built chip designed to facilitate the maintenance and differentiation of tissue-specific spheroids and organoids. Dotted lines represent the potential path an electromagnet-modified 3D printer could take to create an assembloid.
**b** Precision, in X and Y dimensions, of automated alginate microgel transfer. Each data point represents a single microgel wherein each microgel yielded data in both X and Y dimensions ($n = 10$). **c** Drift (i.e., the distance between where an alginate microgel was intended to be deposited and where the microgel settled) in the Z dimension as a function of microgel diameter. Each data point represents a single microgel ($n = 17$). Line of best fit: $y = -0.4183x + 1.421$. Coefficient of determination: $r^2 = 0.45$. **d** Positional stability (i.e., Z dimensional drift relative to the initial displacement) over time of alginate microgels over 72 hours ($n = 7$); error bars represent standard deviation. **e** Representative fluorescence image of an eGFP-expressing ventral forebrain neural organoid fused to two mScarlet-expressing dorsal forebrain neural organoids. **f** Representative fluorescence image of two eGFP-expressing ventral forebrain neural organoids, two mScarlet-expressing dorsal forebrain neural organoids, and two non-fluorescent dorsal forebrain neural organoids from distinct hiPSC lines fused in a ring. **g** Representative fluorescence image of an eGFP-expressing ventral forebrain neural organoid fused to three mScarlet-expressing dorsal forebrain neural organoids in a multi-layered pyramid in which the ventral organoid is above the dorsal organoids. **h** Representative immunofluorescence (IF) image of a ventral forebrain neural organoid integrated with a dorsal forebrain neural organoid. **i** Representative IF image of a ventral forebrain neural organoid integrated with a dorsal forebrain neural organoid with regions of higher magnification to illustrate cell migration. Unless otherwise noted, all data points represent distinct biological replicates. Data plotted as mean ± SD.

and long-term movement of a given OBB. Using a spherical alginate microgel as a non-living, static model, we evaluated the initial drift of deposited spheres with similar diameters to neural organoids. Magnetic bioprinting was subject to initial drifts of −0.31 ± 0.85 mm, −0.12 ± 0.43 mm, and 0.59 ± 0.23 mm in X-, Y-, and Z-dimensions, respectively (Fig. 3b and Supplementary Fig. 15a, b). Interestingly, as the diameter of the microgel increased from 1.3 to 2.5 mm, the initial Z drift decreased, suggesting that neural organoids, which tend to exhibit diameters approaching and exceeding 2 mm, may undergo decreased drift as they grow (Fig. 3c). After deposition, the OBBs must remain immobilized within the support scaffold to permit fusion into a cohesive structure. For neural organoids, fusion is consistently observed over the course of 72 h. During this time, the positional movement along the Z-direction was <5% (Fig. 3d) and was consistent across replicates (Supplementary Fig. 15c, d). Additionally, the total XY drift over 72 h was minimal (0.044 ± 0.0021 mm; Supplementary Fig. 15e).

To demonstrate the multi-dimensional spatial control achieved by the SPOT platform, we manually constructed human dorsal-ventral forebrain assembloids from hiPSC lines that constitutively expressed either mScarlet or eGFP. Fusion was successful across a range of shapes that could not be easily fabricated using current approaches. For example, linear three-part assembloids were constructed with a pre-determined OBB sequence, and six individual neural organoids, derived from three hiPSC lines and differentiated into two domains of the forebrain, were arranged and fused to form a ring-like structure (Fig. 3e, f). Due to the positional stability of the OBBs within the support scaffold, biofabrication of multi-layered structures such as pyramids was also possible (Fig. 3g).

The cerebral cortex can be conceived of as a collection of circuits composed of excitatory glutamatergic neurons derived from the dorsal forebrain and inhibitory GABAergic interneurons derived from the ventral forebrain. The migration of these interneurons into the

human cortex occurs throughout fetal and postnatal development and has been implicated in the etiology of various neuropsychiatric disorders[43]. Previous studies have leveraged human iPSC-derived dorsal-ventral assembloids to characterize the impact of genetic mutations associated with autism spectrum disorder on the saltatory migration of interneurons[8]. Here, we observed the robust integration of iPSC-derived ventral forebrain organoids to dorsal forebrain organoids following their controlled spatial positioning within, and subsequent release from, a CNF support scaffold by magnetic bioprinting (Fig. 3h). Over the course of two weeks post-release from the CNF support scaffold, we observed extensive migration of GABAergic interneurons from the ventral region into the dorsal forebrain region of the assembloids (Fig. 3i, Supplementary Movie 3). These migratory cells exhibited highly branched projections (Supplementary Movie 4) that spanned across a Z-depth as wide as 25 μm (Supplementary Movie 5). In conclusion, regionalized hiPSC-derived neural organoids can be controllably positioned in 3D and exhibit cellular migration indicative of functional integration. Taken together, these observations lay the foundation for the use of SPOT as a platform for constructing complex neural circuits in vitro.

## Bioprinting patient-derived glioma assembloids to study tumor progression and drug response

Diffuse intrinsic pontine glioma (DIPG) is a universally fatal pediatric cancer that arises in the ventral pons[44,45] and exhibits key molecular and genomic differences compared to adult high-grade gliomas[46–48]. While pontine in origin, DIPG has been shown to infiltrate extensively throughout the brain, from the subventricular zone (SVZ) through the frontal cortex[49–51]. Standardized protocols[52] have facilitated the use of patient-derived models that have helped identify promising therapeutic agents[53–55]. However, to date, no experimental models have recapitulated the interactions between DIPG and healthy human neural tissue from distinct brain regions. To demonstrate the potential for SPOT to facilitate studies characterizing the interactions between glioma and human neural tissue ex vivo, we created assembloids consisting of hiPSC-derived regionalized neural organoids harboring distinct SVZ-like regions and patient-derived DIPG organoids with different metastatic profiles. In addition to probing the infiltration of DIPG, we were interested in studying whether a leading drug candidate, panobinostat, might have altered efficacy in the presence or absence of healthy neural tissue from distinct brain regions.

The magnetic bioprinting-enabled localization and subsequent fusion of neural organoids with DIPG organoids was reproducible and scalable (Fig. 4a). As with the multi-region assembloids constructed with SPOT, neuro-DIPG assembloids remained intact after cellulase-mediated release from the CNF support bath. Importantly, the spatial control imparted by the SPOT platform allowed for the creation of assembloids wherein forebrain organoids were integrated with DIPG organoids derived from two distinct brain regions of a single patient: 1) the tumor origination site, the pons (DIPGXIII-P), and 2) a distant brain region, the frontal lobe (DIPGXIII-FL), into which the tumor metastasized[56]. As a proof of principle demonstration, we created an assembloid consisting of a pontine DIPG organoid, a dorsal forebrain neural organoid, and a frontal lobe DIPG organoid (Fig. 4b). The substitution to methionine in histone H3 at lysine 27 (H3K27M), a hallmark of diffuse midline pediatric gliomas[57,58], was predominantly observed within the tumor organoid, and robust infiltration of GFP-expressing DIPG projections was observed at the tissue interface one-week post-fusion. Similar three-part assembloids were created by integrating dorsal and ventral forebrain organoids with a frontal lobe DIPG organoid (Supplementary Fig. 16). These three-part assembloids have the potential to serve as unique tools for investigating tumor infiltration and drug response across a range of therapeutically relevant variables including tumor metastatic state and brain region. Tumor infiltration, in both two-part and three-part assembloids, was characterized by the

migration of H3K27M-expressing DIPG nuclei into the periphery of the neural organoid and by the extension of GFP-expressing projections deeper into the neural organoid throughout networks of neuronal and glial cells (Supplementary Fig. 17, 18). Previous studies of DIPG metastasis have relied upon either patient-derived orthotopic xenografts or genetically engineered mouse models[59,60]; in comparison, these assembloids serve as entirely human, ex vivo models of DIPG infiltration into neural tissue.

To illustrate the utility of the SPOT platform in translational studies, we created an array of neuro-DIPG assembloids. These assembloids included multiple permutations of DIPG progression (i.e., originating pons and metastatic frontal lobe) and multiple neural organoid types (i.e., dorsal and ventral forebrain). Subsequently, we treated the assembloids with panobinostat, a multiple histone deacetylase (HDAC) inhibitor that has recently been identified as a potential therapeutic for DIPG[53,55] and is currently in several clinical trials (NCT02717455, NCT04341311, NCT04804709, and NCT05009992; Supplementary Fig. 19).

As panobinostat was shown to decrease DIPG viability in vitro[53,55], we characterized the expression of the apoptosis marker cleaved caspase-3 in DIPG organoids following assembly and after panobinostat treatment (Fig. 4c). All DIPG organoids treated with panobinostat expressed higher levels of cleaved caspase-3 compared to untreated controls. Moreover, the adjacent neural organoids did not exhibit substantially increased cleaved caspase-3 expression following panobinostat treatment (Supplementary Fig. 20). No statistically significant differences were observed between pontine and frontal lobe DIPG organoids in isolation ($p = 0.996$); however, when fused to either dorsal or ventral neural organoids, pontine DIPG organoids expressed significantly higher levels of cleaved caspase-3 compared to frontal lobe DIPG organoids (ventral fusion: $p < 0.0001$, dorsal fusion: $p < 0.01$; Fig. 4d). This suggests that panobinostat may induce greater degrees of apoptosis in DIPG cells that have not yet metastasized. Critically, this difference was only observable in assembloids, as opposed to the isolated DIPG organoids, highlighting the need for recapitulating cell-cell interactions.

The H3K27M mutation perturbs polycomb repressive complex 2 resulting in global hypomethylation of K27 and DIPG oncogenesis[61–68]. As an HDAC inhibitor, panobinostat rescues the hypotrimethylation phenotype, which should specifically target DIPG cells with the H3K27M mutation. Given panobinostat's proposed mechanism of action and the observed differences in cleaved caspase-3 expression, we hypothesized that panobinostat treatment may deplete H3K27M populations within pontine DIPG organoids to a greater extent than frontal lobe DIPG organoids when fused to neural organoids. Accordingly, the H3K27M expression within isolated DIPG organoids was decreased following panobinostat treatment; however, it did not exhibit significant differences as a function of their metastatic profile ($p = 0.983$; Fig. 4e, f). Conversely, H3K27M expression within assembled pontine DIPG organoids was significantly lower than that within assembled frontal lobe DIPG organoids following panobinostat treatment (ventral fusion: $p < 0.05$, dorsal fusion: $p < 0.01$). Taken together, these data suggest that panobinostat drives a disproportionate loss of DIPG cells with the H3K37M mutation in assembloids wherein the tumor organoid was derived from the originating tumor site. Further studies may be able to leverage SPOT as a platform for exploring additional brain region- or tumor-specific therapeutic models, such as those for pediatric glioblastoma, adult glioblastoma, or anaplastic oligodendroglioma (Supplementary Fig. 21).

## Discussion

Collective tissue behaviors, ranging from morphogenesis to tumor infiltration, are dependent upon cell-cell and cell-microenvironment interactions[69]. These processes are starting to be modeled in self-organizing organoid and assembloid models[1]. However, as we move

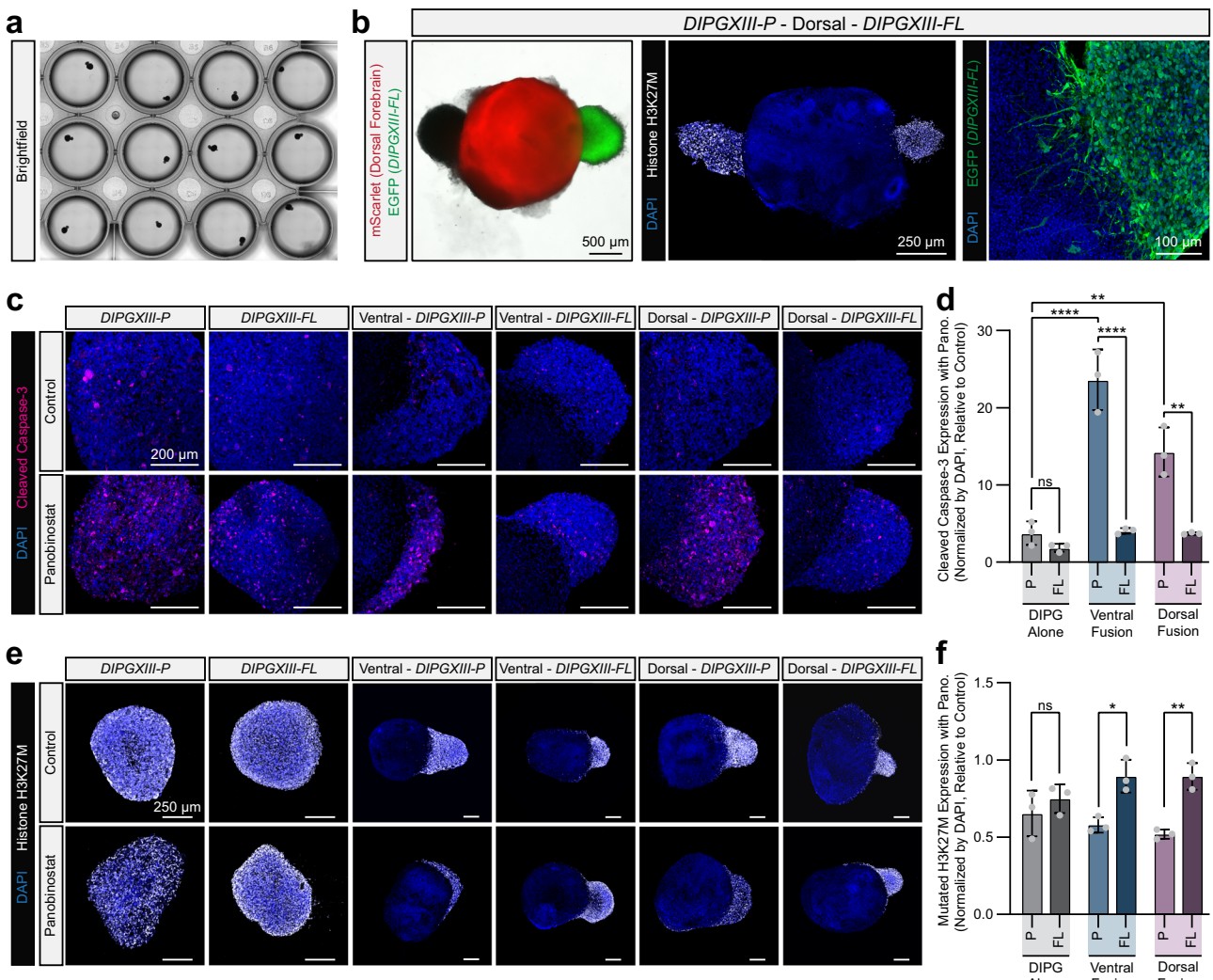

**Fig. 4 | Glioma assembloids predict tumor progression-specific drug response. a** Representative image of a well plate with hiPSC-derived neural organoids fused to patient-derived DIPG organoids. **b** Representative BF and IF images of a three-part assembloid in which two distinct DIPG organoids, derived from either the pons, which is the tumor origination site (DIPGXIII-P), or the frontal lobe, a brain region into which the tumor metastasized (DIPGXIII-FL), are fused to a dorsal forebrain neural organoid. **c** Representative IF staining of the apoptosis marker cleaved caspase-3 across an array of permutations of DIPG organoids fused to neural organoids in which a subset of the assembloids were treated with 200 nM pano-binostat. **d** Quantification of the relative degree of apoptosis as determined by cleaved caspase-3 staining within DIPG organoids normalized by DAPI and relative to the untreated control organoids. Each data point represents a different DIPG organoid either alone or within an assembloid (DIPGXIII-P $n = 3$, DIPGXIII-FL $n = 3$, Ventral-DIPGXIII-P $n = 3$, Ventral-DIPGXIII-FL $n = 3$, Dorsal-DIPGXIII-P $n = 3$, Dorsal-DIPGXIII-FL $n = 3$). $p$ values for shown apoptosis comparisons are as follows (additional $p$ values listed in Supplementary Data 1): Pontine:DIPG Only vs. Frontal Lobe:DIPG Only $p = 0.9958$, Pontine:DIPG Only vs. Pontine:Ventral Fusion

$p < 0.0001$, Pontine:DIPG Only vs. Pontine:Dorsal Fusion $p = 0.001$, Pontine:Ventral Fusion vs. Frontal Lobe:Ventral Fusion $p < 0.0001$, Pontine:Dorsal Fusion vs. Frontal Lobe:Dorsal Fusion $p = 0.001$. **e** Representative IF staining of H3K27M across an array of permutations of DIPG organoids fused to neural organoids in which a subset of the assembloids were treated with 200 nM panobinostat. **f** Quantification of the relative number of H3K27M-expressing cells. H3K27M staining within DIPG organoids was normalized by DAPI and shown relative to the untreated control organoids. Each data point represents a different DIPG organoid either alone or within an assembloid (DIPGXIII-P $n = 3$, DIPGXIII-FL $n = 3$, Ventral-DIPGXIII-P $n = 3$, Ventral-DIPGXIII-FL $n = 3$, Dorsal-DIPGXIII-P $n = 3$, Dorsal-DIPGXIII-FL $n = 3$). $p$ values for shown H3K27M comparisons are as follows (additional $p$ values listed in Supplementary Data 1): Pontine:DIPG Only vs. Frontal Lobe:DIPG Only $p = 0.983$, Pontine:Ventral Fusion vs. Frontal Lobe:Ventral Fusion $p = 0.0206$, Pontine:Dorsal Fusion vs. Frontal Lobe:Dorsal Fusion $p = 0.0055$. Statistical analyses performed as two-way ANOVA with Šídák's multiple comparisons test. Unless otherwise noted, all data points represent distinct biological replicates. Data plotted as mean ± SD where *$p < 0.05$, **$p < 0.01$, ***$p < 0.001$, ****$p < 0.0001$, and ns not significant.

towards recapitulating increasingly complex, multi-lineage interactions in vitro, synergizing advances in OBB creation with innovations in biofabrication will prove to be critical[36]. Here, we develop an organoid bioprinting platform, termed SPOT, wherein individual OBBs can be positioned in 3D space with both a high degree of spatial control and the preservation of internal cytoarchitecture. The positioning of these OBBs is achieved through the use of an MNP-laden, bioinert hydrogel that envelops the tissue of interest and facilitates electromagnet-mediated lifting, transfer, and deposition within a hydrogel support scaffold. Within this matrix, OBBs can undergo fusion to create

assembloids. With SPOT, we construct neural assembloids to serve as in vitro models both of a neurodevelopmental phenomenon, namely the migration and integration of interneurons into the pallium, and of neural disease progression, namely the infiltration of tumor cells into distinct brain regions.

This magnetic bioprinting approach is inspired by previously reported pick-and-place biofabrication techniques, namely AAB[28–31]. Compared to vacuum aspiration mediated OBB printing, SPOT reduces the concentrated localization of force on the tissue surface and is therefore uniquely suited for OBBs with low resistance to deformation

and applications in which the cytoarchitecture of the OBB is relevant to the physiology of interest. Additionally, whereas AAB is predicated upon manual selection of OBBs within a media reservoir, SPOT utilizes a custom chip design with microwells for each OBB. This allows for the potential use of G-code to automate locating, lifting, and depositing the OBBs at a specified position within the support bath. It should be noted that the fusion of OBBs has also been previously achieved with the Kenzan method in which an OBB is aspirated, impaled with a metal microneedle, and, over the course of multiple such piercings with additional OBBs, fused into a single assembloid[70]. While this approach has been automated and commercialized, its dependency on puncturing the OBB and the concomitant deformation of the punctured OBB severely undermines its use for OBBs with conserved, biologically relevant cytoarchitecture. Moreover, this approach is limited in the complexity of OBB configurations it can create given the rigidity of the needles. Taken together, when compared to other OBB printing technologies, SPOT is a substantial improvement as it introduces spatial fidelity in 3D without damaging the constitutive OBB.

Magnetic forces have been previously shown to mediate the formation of patterned 3D tissues from single cells in a now commercialized process known as magnetic levitation[71,72]. While magnetic levitation and SPOT both rely on MNPs, there are several key differences between the platforms. Firstly, with magnetic levitation, individual cells are maneuvered into a desired geometry. SPOT mediates the controlled movement of entire spheroids or organoids and is therefore uniquely suited to applications wherein the cytoarchitecture of an OBB is critical to its fidelity as a model. Secondly, magnetic levitation is predicated on the cellular uptake of a bioinorganic hydrogel containing iron oxide, while SPOT temporarily coats the surface of an OBB with an MNP-laden hydrogel. This transient exposure to MNPs limits the potential for OBBs to undergo any MNP-induced alterations in cellular phenotype. Therefore, when compared to this previous magnetic bioprinting approach, SPOT is particularly well suited for constructing assembloids from organoids with conserved cellular arrangements.

SPOT aims to serve as a complementary approach to conventional assembloid formation protocols that are dependent on the fusion of organoids due to confinement within a microcentrifuge tube[16]. While these protocols rely upon reagents and equipment readily available in most biology laboratories, the simplicity of the assembly itself limits the degree of control imparted over the spatial positioning of the OBB. Moreover, while linear assembloids composed of up to three distinct OBBs have been demonstrated[12], building assembloids in X, Y, and Z dimensions remains a challenge. As such, when compared to current state of the art OBB assembly protocols, SPOT has the potential to serve as an improvement insofar as the electromagnet-modified 3D printer allows users to control the positioning of multiple OBBs in three dimensions.

We engineered the SPOT platform to be accurate, scalable, and readily adoptable, and have identified several technical steps that may be of interest to those intending to incorporate it into their experimental workflows. First, the MNP concentration, coating time, magnetic rod diameter, and magnetic field strength must be optimized for the largest OBB in an experiment. Second, while SPOT can accommodate a range of OBB diameters from 300–3000 μm, it struggles to accurately deposit OBBs under 300 μm. Further optimization of the attachment of such organoids to the magnetic rod may ameliorate this particular limitation.

Finally, as this bioprinting platform is OBB-agnostic, it can be utilized across a wide range of biological systems wherein signaling from distinct cell types, lineages, and oncogenic potential is relevant. Here, we leverage SPOT to construct multi-region neural assembloids consisting of regionalized constituents of neural circuits and tumor-host assembloids wherein the ratio and positioning of each OBB is controllably varied. We envision that combination of the SPOT platform with spatially-resolved single-cell RNA sequencing, multi-plexed time-lapse immunofluorescence, and imaging mass cytometry would have the potential to reveal compelling mechanistic insights into the spatiotemporal dynamics of tumor infiltration. Future studies may adopt the platform for investigations into the developmental trajectory of various tissues or etiology of various diseases and, in so doing, mediate the discovery and preclinical validation of therapeutics.

## Methods

### hiPSC maintenance
The stemness and differentiation capacity of the human induced pluripotent stem cells (hiPSCs) used in this study were previously validated[73,74]. All hiPSCs were tested for and maintained mycoplasma free. In total, 4 hiPSC lines from two distinct donors were included. Approval for this study was obtained from the Stanford IRB, and informed consent was obtained from all donors.

hiPSCs were maintained using standard methods. Briefly, hiPSCs were cultured with mTESR-1 Plus (StemCell Tech 100-0276) media in monolayer on hESC-qualified Matrigel (Sigma 354277).

### Neural organoid differentiation and maturation
Dorsal and ventral forebrain neural organoids were differentiated in accordance with previously published protocols[8,16,75]. For both brain regions, hiPSCs were dissociated with Accutase (StemCell Tech 07920), aggregated into uniform 5000 cell aggregates with Aggre-Well800 plates (StemCell Tech 34815), and allowed to stabilize for 16 hours (h) in mTeSR-1 Plus with ROCK inhibitor Y-27632 (10 μM, StemCell Tech 72307). hiPSC aggregates were then transferred to ultralow-attachment plastic dishes (Corning 3471) with hiPSC media consisting of Essential 6 medium (Gibco A1516401) supplemented with penicillin-streptomycin (1:100, Gibco 15140122).

For dorsal brain region specific organoids, hiPSC media was additionally supplemented with the two dual SMAD inhibitors LDN-193189 (100 nM, StemCell Tech 72147) and SB-431542 (10 μM, Tocris 1614) and changed daily. On the sixth day in suspension, hiPSC medium was replaced with neural medium consisting of neurobasal-A (Thermo Fisher Scientific 10888022), B-27 supplement without vitamin A (1:50, Thermo Fisher Scientific 12587010), GlutaMax (1:100, Thermo Fisher Scientific), penicillin-streptomycin (1:100, Gibco 15140122), and supplemented with human EGF (20 ng ml$^{-1}$, PeproTech AF-100-15) and human FGF-2 (20 ng ml$^{-1}$, PeproTech AF-100-18B) through day 24. From day 25 to 42, neural medium was supplemented with the growth factors BDNF (20 ng ml$^{-1}$, PeproTech AF-450-02) and NT3 (20 ng ml$^{-1}$, PeproTech AF-450-03) with medium changes every other day. From day 43 onward, dorsal neural organoids were maintained in neural medium with medium changes every four days.

For ventral brain region specific organoids, hiPSCs were differentiated following the same protocol described for dorsal neural organoids with two important amendments. Firstly, from day 4 to day 24, the WNT pathway inhibitor IWP2 (5 μM, Selleckchem S7085) was added. Secondly, from day 12 to day 24, the SHH pathway agonist SAG (100 nM, Selleckchem S7779) was added.

Neural assembloids fused within four days. Throughout their culture, assembloids were maintained in neural medium with medium changes every four days.

### MSC and HUVEC spheroid culture
Human MSCs (Lonza PT-2501) were expanded in high-glucose DMEM with GlutaMAX (Thermo Fisher Scientific 10566016) supplemented with FBS (1:10, Thermo Fisher Scientific 12662029) and penicillin-streptomycin (1:100, Gibco 15140122).

Human umbilical vein endothelial cells (PromoCell C-12200) were expanded in endothelial growth medium-2 (EGM-2 bullet kit, Lonza CC-3162).

To form uniform sized spheroids, MSCs and HUVECs were dissociated, aggregated as either 5000 or 8000 cell clusters, respectively, using AggreWell800 plates, and allowed to stabilize for 16 hours. Spheroid formation was confirmed by phase contrast microscopy and maintained with daily media changes.

## Primary brain tumor organoid culture
Patient-derived primary cells (SU-DIPG-XIII-FL, SU-DIPG-XIII-P, pcGBM-2, GBM-81, SU-AO-3) were provided by the lab of Prof. Michelle Monje-Deisseroth (Stanford University). All human tumor cell cultures were generated with informed consent and under institutional review board (IRB)-approved protocols, as previously described[53,56,76]. Tumor cells were expanded as tumor neurospheres in tumor stem medium consisting of neurobasal (Thermo Fisher Scientific 21103049), B-27 supplement without vitamin A (1:50, Thermo Fisher Scientific 12587010), human EGF (20 ng ml$^{-1}$, Shenandoah Biotech 100-26), human b-FGF (20 ng ml$^{-1}$, Shenandoah Biotech 100-146), human PDGF-AA (10 ng ml$^{-1}$, Shenandoah Biotech 100-16), human PDGF-BB (10 ng ml$^{-1}$, Shenandoah Biotech 100-18), and heparin (2 ng ml$^{-1}$, StemCell Tech 07980). Media was changed once per week.

## Organoid and spheroid mass, diameter, and apparent surface tension measurements
The following characterizations were performed similarly for neural organoids, MSC spheroids, and HUVEC spheroids; to simplify the description of the methods, all three structures are broadly referred to as OBBs.

To measure mass, OBBs were manually transferred to an Eppendorf tube containing a small volume of DPBS (Corning 21-031-CM) by first pipetting an individual OBB onto the edge of a metal spatula, manually removing any excess media, then lightly tapping the OBB to the surface of the DPBS. This process was repeated four additional times such that the Eppendorf tube contained five spheroids or organoids. The resultant mass was divided by five to obtain a single, averaged data point. This process was then repeated five times per spheroid type and time point.

To measure diameter, brightfield images were recorded with an epifluorescent microscope (Leica Microsystems, THUNDER Imager 3D Cell Culture) and the diameter was manually traced using ImageJ (NIH, v.2.1.0/1.53c).

To measure apparent surface tension, OBBs were exposed to micropipette aspiration as previously described[28,77,78]. Briefly, a clear, plastic, blunt edge nozzle on a syringe was affixed to a DBPS-containing 35 mm plate. The syringe was connected to a pressure modulator, which was connected to a vacuum line. A range of pressures ($\Delta P = 1-30$ mmHg) was applied to the OBB surface, and the subsequent deformation was observed on an epifluorescent microscope (Leica Microsystems, THUNDER Imager 3D Cell Culture).

## Vacuum aspiration
OBBs were exposed to vacuum aspiration pressures following the same protocol described above for apparent surface tension measurements. Importantly, to ensure consistent pressure was applied to a given OBB, the desired pressure was first reproducibly obtained in DPBS before the blunt edge nozzle was lowered to the surface of the OBB.

## Neural organoid viability
To characterize viability, organoids were submerged in a solution consisting of DPBS supplemented with 2 µM calcein AM and 4 µM ethidium homodimer for 20 min at 37 °C (Thermo Fisher Scientific L3224). The samples were washed with DPBS and imaged with an epifluorescent microscope (Leica Microsystems, THUNDER Imager 3D Cell Culture) and confocal microscope (Leica SPE).

## MNP fabrication
Iron oxide magnetic particles were fabricated in-house by the conventional method of co-precipitation, in which ferrous and ferric ions are mixed in a 1:2 molar ratio in a basic solution[79]. Briefly, 0.05 M iron(II)sulfate (Sigma-Aldrich F7002) and 0.1 M iron(III)chloride (Sigma-Aldrich 157740) were first dissolved together in water at room temperature. A solution of 10% ammonium hydroxide was added to the reaction dropwise through a separatory funnel with constant stirring (500 rpm) for 1 h. Following the completion of the reaction, the iron oxide particles were washed three times with water.

Commercial iron oxide nanoparticles were purchased from Alpha Nanotech Inc. (size: 300 nm, surface coating: polydopamine coating).

## MNP size distribution and zeta potential
To determine the hydrodynamic size distribution, dynamic light scattering was performed on a 0.1 wt% MNP in DPBS solution using a Malvern Zetasizer Nano ZS. To determine the distribution of aggregate sizes, a 0.1 wt% MNP in DPBS solution was sonicated in a bath sonicator for 5 min, sandwiched between two glass slides, imaged with a Leica THUNDER microscope, and processed with FIJI. To determine the zeta potential, a 0.01 wt% MNP in DPBS solution was sonicated in a bath sonicator for 5 minutes, suspended in folded capillary cells (Malvern DTS1060), and characterized, with 10 runs per sample, in a Malvern Zetasizer Nano ZS.

## MNP surface coverage
To quantify organoid surface MNP coverage, brightfield Z-stack images were taken using a Leica THUNDER microscope. Background subtraction was performed using the rolling ball algorithm (radius = 25 pixels), and MNP coverage area was measured via thresholding and maximum Z projection.

## Magnetic field strength characterization
To characterize the magnetic field applied during lifting, magnetic field strength measurements were performed as a function of applied voltage and distance from the probe tip using a LATNEX MK-30K AC/DC Gauss meter and electromagnet-modified Prusa i3 MK3S 3D printer.

## CNF fabrication
Stock solutions of bacterial CNF for the magnetic ink and support scaffold were fabricated from nata de coco (Jubes). The nata de coco cubes were washed with flowing deionized water for two days. Following the washes, coco de nata and deionized water were blended together in a 1:1 ratio until homogenous. The solution was concentrated through centrifugation at 12857×g for 20 minutes, autoclaved for sterilization, and stored at 4 °C. To calculate the concentration of the CNF stock solution, aliquots were weighed before and after drying. For use in the magnetic ink or support scaffold, the CNF stock was diluted with sterile DPBS.

## CNF macrorheological characterization
Mechanical testing of the CNF-based magnetic ink and support scaffold formulations was performed using an AR-G2 (TA Instruments) stress-controlled rheometer (8 mm and 40 mm parallel plate geometry with a 1 mm gap) at 25 °C. For the storage and loss moduli, frequency sweeps were performed between 0.1 and 100 rad s$^{-1}$ at a strain of 1%, and measurements were confirmed to be within the linear viscoelastic regime. Viscosity tests were performed at shear rates ranging from 0.1 to 10 s$^{-1}$. For self-healing measurements, alternating strains of 0.1% and 300% were applied.

## Magnetic lifting
OBBs were coated with 10 µL of a magnetic ink composed of 1 wt% MNPs in 0.025 wt% CNF for 30 min. Magnetic rods were affixed to

an electromagnet set to 15 V. The rod was then lowered until it was just above the surface of the coated OBB. Once the OBB attached to the rod, the entire construct was moved throughout DPBS to emulate the movement an OBB would experience in the printing process.

## Bioprinting chip design and fabrication
To facilitate automated magnetic bioprinting, we designed a chip that ensured that a series of OBBs were consistently located at a given position. Additional features included an offset platform for medium addition, a row of elongated U-bottom wells, and a raised connector channel between the wells and the reservoir which contained the support scaffold. The chip was created by pouring an uncured mixture of Sylgard 184 (Dow Corning 2646340) in 10:1 base to curing agent ratio into a 3D printed polylactic acid mold. The PDMS was degassed under vacuum for 20 min and cured at room temperature (RT) for 48 h before being carefully removed from the mold.

## Automated magnetic bioprinting
Automated transfer was achieved using an electromagnet-modified Monoprice MP Select Mini 3D Printer V2 and Kaiweets PS-3010F DC power supply. The printer was modified such that the print-head fan controls were wired, via a solid-state relay, to the power supply which, in turn, activated an electromagnet. As the fan can be turned on or off with G-Code, the electromagnet itself was controllable with the same code that was used to address the movement of the OBB.

## Precision of automated magnetic bioprinting
For the measurement of XY localization, alginate beads with diameters comparable to those of neural organoids (1–2 mm) were coated with a magnetic ink composed of 1 wt% MNPs in 0.025 wt% CNF for 30 min and transferred from water to a pre-specified location within a 0.5 wt% CNF bath using the magnetic 3D bioprinting approach. ImageJ (NIH, v.2.3.0/1.53q) was used to measure the XY deviation of the deposited alginate bead from the marked position.

To measure bioprinting precision in Z, alginate beads were transferred from water into a 0.5 wt% CNF support bath with the magnetic rod. The Z position of the bead was tracked via imaging over 3 days.

## CNF support diffusivity characterization
The diffusivity within the CNF support scaffold was assessed by fluorescent recovery after photobleaching (FRAP) measurements. Briefly, 0.5 wt% CNF was prepared with encapsulated FITC-dextran probes (Sigma) with molecular weights of 10 kDa, 20 kDa, 40 kDa, 70 kDa, 150 kDa, 250 kDa, and 500 kDa. The FRAP experiments were then performed on a confocal microscope (Leica SPE) with 1 min of photobleaching (100 μm x 100 μm bleach area, 488 nm laser, 100% intensity) followed by 4 min of capture time (10% intensity). The diffusion coefficients for each probe size were determined using the open-source MATLAB code "frap_analysis" based on the Hankel transform method[80].

## Cellulase-mediated CNF scaffold degradation
For measurements of cellulase-mediated CNF support scaffold degradation, cellulase (Sigma-Aldrich C1794) was dissolved in DMEM/F12 (Thermo Fisher Scientific 11320033) and added atop 0.5 wt% CNF in a 1:4 ratio by volume to approximate the ratio of media to CNF support within the bioprinting chip. All samples were incubated in a humidified 37 °C incubator for 3 days. Depending on the downstream measurement, cellulase solutions, as well as non-cellulase containing DMEM/F12 controls, were either changed every day or allowed to incubate over the full three days without a media change.

Viscosity and storage moduli were obtained with an AR-G2 (TA Instruments) stress-controlled rheometer (20 mm 1° cone and plate geometry with a 28 μm gap). To observe the effect of cellulase on the viscosity of CNF over 72 h, samples were loaded onto the stage at 37 °C for a 5 min time sweep with 1% oscillatory strain and 1 rad/s angular frequency. This was followed by a frequency sweep from 0.1 to 100 Hz at 1% strain. To measure the storage modulus, samples were loaded onto the stage at 2 h, 4 h, 6 h, 24 h, 48 h, and 72 h and subjected to a frequency sweep at 1 rad/s angular frequency.

## Cellulase-mediated degradation of CNF surrounding extracted neural organoids
To characterize cellulase-mediated CNF degradation after neural organoid release from the support scaffold, neural organoids were first cultured in 0.5 wt% CNF support bath for 1 day and released through diluting the support bath with DPBS. A 0.5 wt% cellulase (Sigma) solution, made up in neural medium, was filtered through a 0.22 μm filter, warmed to 37 °C, and added to the organoids with daily medium changes. Organoids were imaged every day with an epifluorescent microscope (Leica Microsystems, THUNDER Imager 3D Cell Culture). The area of residual CNF on the organoid surface was manually measured with ImageJ (NIH, v.2.3.0/1.53q).

## Quantitative reverse transcription polymerase chain reaction
mRNA expression was quantified with quantitative reverse transcription polymerase chain reaction. Organoids were suspended in 500 μL of TRIzol reagent (Thermo Fisher Scientific 15596026) and disrupted via probe sonication (Heilscher UP50H, 50% amplitude (25 watts), 30 kHz frequency, 0.5 cycle). mRNA was purified by phenol-chloroform extraction with phase lock gels (Quantabio 5PRIME 2302830) followed by isopropyl alcohol precipitation. The resultant mRNA was resuspended in nuclease-free water (Thermo Fisher Scientific 10977015) and measured via NanoDrop (Thermo Fisher Scientific). 100 ng of mRNA was reverse transcribed using a High-Capacity cDNA Reverse Transcription Kit (Applied Biosystems 4368814). For qPCR, 6.6 μL of diluted cDNA was mixed with 0.9 μL of a 5 μM forward and reverse primer pair (Integrated DNA Technologies, Supplementary Table 3) solution and 7.5 μL of Fast SYBR Green Master Mix (Applied Biosystems 4385612). Samples were run on a StepOnePlus Real Time PCR System (Applied Biosystems). CT values were calculated using the StepOnePlus software (v.2.3) and analyzed by the ΔCT method.

## Panobinostat treatment of neuro-DIPG assembloids
Neuro-DIPG assembloids fabricated with SPOT fused within 24 h. They were subsequently released from the CNF support bath using cellulase treatment as described above and cultured for one week in suspension, with media changes performed every 3-4 days. After 1 week in suspension culture, the media was replaced with fresh media containing 200 nM panobinostat (Selleckchem S1030). Neuro-DIPG assembloids were cultured in the presence of 200 nM panobinostat for 72 h, with no media changes, after which samples were fixed in 4% paraformaldehyde (PFA, Electron Microscopy Sciences 15700) in DPBS for immunohistochemistry. Control samples were also given fresh media without panobinostat on day 7 and cultured for 72 h.

## Immunohistochemistry
Organoids and assembloids were fixed in 4% PFA for 2 h at 4 °C. They were then washed three times with DPBS, for 15 min each, and transferred to a 30% sucrose solution in DPBS for 24–48 h at 4 °C. Once the organoids or assembloids sank in the sucrose solution, they were embedded in a 1:1 mixture of OCT (Fisher Scientific 23-730-571) and 30% sucrose in DPBS. They were then snap frozen on dry ice and stored at −80 °C. A cryostat (Leica) was used to cut 50 μm sections for immunostaining.

Cryosections were washed with DPBS to remove excess OCT, then permeabilized with 0.25% Triton X-100 (Thermo Fisher

Scientific A16046) in DPBS (DPBS-T) for 1 h and blocked with 5% goat serum (Gibco 16210-072), 5% bovine serum albumin (BSA, Sigma A9418), and 0.5% Triton X-100 in DPBS for 3 hours, all at RT. Samples were stained with primary antibodies for GFP, (1:200, Thermo Fisher Scientific a11122), cleaved caspase-3 (1:400, Cell Signaling 9661), histone H3 mutated K27M (1:400, Abcam ab190631), paired box 6 (1:200, Biolegend 901301), NK2 homeobox 1 (1:100, Thermo Fisher Scientific ma5-13961), beta-tubulin 3 (1:500, Aves Labs TUJ), and GFAP (1:500, Aves Labs GFAP). Primary antibodies were diluted in 2.5% goat serum, 2.5% BSA, and 0.5% Triton X-100 in DPBS and incubated with the samples overnight at 4 °C. Next, the samples were washed with DPBS-T ($3 \times 30$ min, RT) and incubated with secondary antibodies Alexa Fluor 488 (1:500, Thermo Fisher Scientific A-11034), Alexa Fluor 594 (1:500, Thermo Fisher Scientific A-11020), Alexa Fluor 633 (1:500, Thermo Fisher Scientific A-21103), and 4′,6-diamidino-2-phenylindole (DAPI, 5 mg/mL stock, 1:2000, Thermo Fisher Scientific 62247) in the same antibody dilution solution overnight at 4 °C. Finally, the samples were washed with DPBS-T ($3 \times 20$ min, RT) and mounted to No. 1 glass cover slips with ProLong Gold Antifade Reagent (Cell Signaling 9071). Stained samples were imaged using a confocal microscope (Leica SPE) and process with Las-X software (Leica).

### Image analysis

Cleaved caspase-3 expression and H3K27M expression were analyzed from maximum projection immunofluorescence images using CellProfiler[81]. Images were cropped such that only the DIPG organoid area was included in analysis. For each cell, nuclei and H3K27M objects were identified using the "IdentifyPrimaryObjects" command with a "Minimum Cross-Entropy" thresholding method. Cleaved caspase-3 objects were identified using the "IdentifyPrimaryObjects" command with an "Otsu" thresholding method. Overall area of expression was obtained using the "MeasureImageAreaOccupied" command. Cleaved caspase-3 expression and H3K27M expression were normalized by DAPI count and then normalized to the untreated control.

### Statistical analysis and reproducibility

Statistical analyses for this study were performed using GraphPad Prism v.9.3.1 software. Details of specific statistical methods and p-value results are included within the figure captions and summarized in Supplementary Data 1. For all studies, ns = not significant ($p > 0.05$), *$p < 0.05$, **$p < 0.01$, ***$p < 0.001$, ****$p < 0.0001$.

All representative images of neural organoids were obtained from four independent differentiation experiments with similar results. Images associated with aspiration, deformation, and organoid lifting were obtained from at least four independent repetitions. Images associated with organoid printing, fusion, and integration were obtained from at least ten independent repetitions. Images associated with tumor infiltration and drug treatment were obtained from at least three independent repetitions.

### Reporting summary

Further information on research design is available in the Nature Portfolio Reporting Summary linked to this article.

## Data availability

All data supporting the results reported in this manuscript are available in the Stanford Digital Repository under the persistent https://purl.stanford.edu/sw198jy9339 and the https://doi.org/10.25740/sw198jy9339.

## Code availability

All G-code central to the use of SPOT for organoid bioprinting is included as supplementary information.

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

## Acknowledgements

The authors would like to acknowledge Bauer LeSavage for discussion and neural organoid maintenance, Riley Suhar for assistance with assembloid sectioning and immunohistochemistry, Rameshwar Rao for discussion, Ji-il Kim for assistance with assembloid sectioning, Jonas Fowler, Kyle Loh, Fabian Suchy, Joydeep Bhadury, and Hiro Nakauchi for the derivation and genetic manipulation of the hiPSC lines, Jared Hysinger and Michelle Monje for providing the DIPG cells, and patients and their families for their generous donations.

This work was facilitated by support from the Wu Tsai Neurosciences Brain Organogenesis Project, the National Institutes of Health (R01 EB027171, S.C.H.), and the National Science Foundation (NSF) (DMR 2103812, CBET 2033302, S.C.H.).

J.G.R. acknowledges support from the NSF Graduate Research Fellowship Program (GRFP) (DGE-1656518) and the Stanford Smith Family Graduate Fellowship. L.G.B. acknowledges support from the NSF GRFP (DGE-1656518). M.S.H. acknowledges support from the NSF GRFP (DGE-1656518) and the Stanford ChEM-H O'Leary-Thiry Graduate Fellowship.

## Author contributions

J.G.R. designed the research, conducted experiments, analyzed the data, assembled the data into figures, and wrote the manuscript. L.G.B. assisted with the design and execution of experiments pertaining to biomaterials synthesis and characterization as well as bioprinting accuracy. M.S.H. assisted with the design and execution of experiments pertaining to organoid fusion, bioprinting accuracy, and drug efficacy. Y.L. assisted with the design and execution of experiments pertaining to enzymatic degradation of the biomaterial scaffold. B.C. assisted with the design and execution of experiments pertaining to bioprinting accuracy and organoid coating with nanoparticles. Sa.S. assisted with the design and execution of experiments pertaining to DIPG fusion and drug treatment. Su.S. assisted with the design and execution of experiments pertaining to biomaterials synthesis, characterization, bioprinting accuracy and feasibility, and early efforts to automate the platform. F.Y., S.P.P., and S.C.H. provided guidance on the project and interpretation of data.

## Competing interests

J.G.R., L.G.B., Su.S., and S.C.H. are inventors on a patent application (no. 63/337,794) submitted by the Board of Trustees of Stanford University. The authors declare no other competing interests.
