## [Peer Review File · Nature Communications]

Spatially controlled construction of assembloids using bioprintingREVIEWER COMMENTS

Reviewer #1 (Remarks to the Author):

The authors describe biofabrication approach that they term STAMP (Spheroid Transfer Assisted by Magnetic Printing) to place human pluripotent stem cell derived neural organoids and patient-derived glioma organoids adjacent to one another to create “assembloids”. They evaluate this process and then deploy these assembloids in a brain cancer-oriented drug study.

Overall, the manuscript itself is largely quite well written. The exception are some minor grammatical mistakes and words that were left out. The characterization of the STAMP process is quite well described and characterized. However, this manuscript suffers from a number of problems. First, it is very short. While “Communications” is in the title of Nature Communications, most articles published in Nature Communications are not short communications-style articles, but rather full research articles. Second, this STAMP approach to bioprinting seems like a solution to a problem that doesn’t exist. There are other ways to position spheroids/organoids near one another. A number of papers were published between 2003 and 2010 that did this very thing. The drug screening study is incredibly specific. The choice for using diffuse intrinsic pontine glioma (DPIG) rather than a more common glioma/brain tumor is not clear. As such, the impact of the study is diminished. Lastly, this method does not appear to be high throughput. It seems like a relatively slow process, and thus doesn’t add very much to the papers from the 2000s that, while pioneering bioprinting, did so by a very slow spheroid deposition methodology.

Specific Comments:

1. There are some typos/grammatical errors throughout; Lack of simple words such as “of”, “the”, etc., that should be double checked.
2. The introduction is extremely short. 3D printing is much broader than simply placing spheroids/organoids. Moreover, there are a plethora of bioactive bioinks that have contributed to the field. None of this is discussed as background. There is also only slight discussion of the need for the presented technology in the field of neurology. Nothing in terms of neuro-oncology (gliomas) is presented. Overall, the introduction does a poor job at setting up the rest of the paper and should thoroughly overhauled.
3. In the Results, the authors lead with a comparison of spheroid diameters between types of spheroids at an arbitrary time point (day 25). Of course as the diameter increases, their mass will, and subsequently the minimum pressure required to lift the spheroids using AAB will increase. This is all quite obvious. Why not limit the timeline for neural organoids and print them while they are smaller in size? There is also one of the original bioprinting methods pioneered by the Forgacs lab that uses spheroids printed from microcapillary tubes that gets around the limitations of AAB. This method is also

relatively low throughput, but probably not much more than AAB.

4. Figure 2. The requirement for the support scaffold to be directly adjacent to the wells in which the spheroids/organoids form are a massive limitation in the translational nature of this technology.

5. Figure 3. This figure is reminiscent of data published by the Forgacs, Mironov, Markwald, and Prestwich labs between 2003 and 2009. It's not clear how this new low throughput method of depositing spheroids has basic science or translational impact.

6. Panobinostat is a very specific and best case treatment to use. A more convincing study would screen multiple compounds, including clinically-relevant therapies given that many "mechanistically effective" drugs fail in brain tumors because they cannot cross the blood brain barrier. Moreover, this would better test the scalability of the platform, which while touted as "scalable" seems relatively low throughput.

7. Like the Introduction, the Discussion is quite brief. It largely just describes the STAMP approach. There is essentially no discussion or interpretation of the data presented.

8. Why is nanocellulose the biomaterial of choice? It is not a native extracellular matrix-derived biomaterial. Why not use a material that actually would be compatible with human biology? Bioprintable and shear thinning ECM-based/inspired biomaterials/bioinks exist.

Reviewer #2 (Remarks to the Author):

In this article a new platform called Spheroid Transfer Assisted by Magnetic Printing (STAMP) that aims to improve the biofabrication of 3D tissues by providing temporal and spatial control of cell-cell interactions. The platform consists of an iron-oxide nanoparticle-laden hydrogel and a magnetized 3D printer that can lift, transport and deposit spheroids and organoids. Cellulose nanofibers are used as the biomaterial for encasing organoids with magnetic nanoparticles and as a support hydrogel for maintaining the spatial positioning of organoids to create precisely arranged assembloids. The STAMP platform was used to create assembloids composed of human pluripotent stem cell derived neural organoids and patient-derived glioma organoids. The article is well written, has solid novel idea and supported by robust experimental design and well performed indicating the potential to construct assembloids. I suggest publication of this article with minor revision.

It worth to also discuss the potential challenge of this technique, for example of the authors experienced any challenges with floating of OBBs?, attachment of multiple OBB to magnetic tip, attachment of OBB to each other etc. also the author may comment on movement of the magnetic tip

in Z axis to pick the OBB and how they could define the Z value for the print head considering that OBBs may not have the exact same height or may not be distributed as a single layer or not as a single OBB at the bottom of the container.

The author mentioned that a potential cytocompatible ink should have a zero-shear viscosity to prevent MNP sedimentation and to be shear thinning, however the provided rheological graph particularly for 0.1% CNF doesn't show the viscous behaviour of the material at very low shear to show the zero shear behaviour of the material.

Authors are also encouraged to discuss the byproduct of cellulase activity, e.g. oligosaccharides and its potential effect on the OBB viability and functionality.

It could benefit from more specific examples of the types of multi-region neural assembloids and tumor-host assembloids that can be created using the STAMP platform

In the discussion section the authors mentioned the advantage of STAMP, here it is worth to also discuss the time of the two processes of AAB and STAMP. In addition, the cost of the process considering the time and challenges of the coating process can be also discussed to better justify the position of such technologies.

Reviewer #3 (Remarks to the Author):

Roth et al. developed a novel platform wherein they can spatially control assembloids using magnetic based bio printing. The authors then exploited these techniques to develop clinically relevant glioma models. The results emphasize the development of the novel protocol with less attention on fully characterizing the pre-clinical model. Nonetheless, overall this is an interesting and meaningful contribution to organoid/assembloid engineering. Technical advancements with organotypic assembloids have broad implications in cancer as well as other fields such as tissue engineering.

Minor considerations for the authors to consider are as follows:

1. One of the main appeals of the work is having spatial control. In section 2.6, it is not clear how the protocol aided in high throughput dorsal-ventral differentiation even when evaluating supplementary figure 12. Further description of this key element would be useful as well as inclusion of how dorsal versus ventral was verified with staining etc.
2. The authors should add timepoints for the glioma assembloids. For example in Figure 4 and/or methods as well as results, the following questions need to be answered: how long were assembloids in culture before adding the HDAC inhibitor? What is the temporal course of the invasion of DIPG into the neuro organoids? Was the temporal course of invasion different between the various groups (i.e tumor origin site, or dorsal/ventral region)?

Major considerations for the authors to consider are as follows:

1. DIPG as noted by the authors themselves mainly is in midline structures such as the brainstem. Demonstration of invasion into forebrain by a pontine DIPG does not seem very relevant clinically without a comparison to brainstem. Engineering brainstem neural organoids are likely beyond the scope of this manuscript but can the authors consider utilizing cortical glioma samples such as glioblastoma. The authors do have a DIPG that metastasized to the cortex but this is exceedingly rare and lacks robust clinical relevance. The utilization of an invasive primary glial tumor that infiltrates into the cortex seems better suited for this portion of the paper and can augment the DIPG work.
2. The authors discuss the importance of the assembloids in order to study microenvironmental impacts as well as cell-cell interactions. Figure 15 does show invasion of a DIPG organoid into a dorsal neural organoid. What cell types are interfacing with these invading cells? Were they different in the dorsal versus ventral neural organoids or with the pontine or metastatic DIPG organoids? Further demonstration that the assembloids truly can be used for understanding cell-cell interactions would be insightful with consideration to single cell sequencing, multiplex immunofluorescence, proteomics, or other methods that can help delineate these interactions further.

Response to Reviewers

Manuscript ID: NCOMMS-22-52575

“Spatially controlled construction of assembloids using bioprinting”

We have addressed the reviewers' verbatim comments point-by-point below. Reviewer comments are reproduced verbatim in black, our responses are in blue, and specific changes to the manuscript are in red. The same red font has been applied to new text in the revised manuscript.

Reviewer #1 (Remarks to the Author):

The authors describe biofabrication approach that they term STAMP (Spheroid Transfer Assisted by Magnetic Printing) to place human pluripotent stem cell derived neural organoids and patient-derived glioma organoids adjacent to one another to create “assembloids”. They evaluate this process and then deploy these assembloids in a brain cancer-oriented drug study.

Overall, the manuscript itself is largely quite well written. The exception are some minor grammatical mistakes and words that were left out. The characterization of the STAMP process is quite well described and characterized. However, this manuscript suffers from a number of problems. First, it is very short. While “Communications” is in the title of Nature Communications, most articles published in Nature Communications are not short communications-style articles, but rather full research articles. Second, this STAMP approach to bioprinting seems like a solution to a problem that doesn't exist. There are other ways to position spheroids/organoids near one another. A number of papers were published between 2003 and 2010 that did this very thing. The drug screening study is incredibly specific. The choice for using diffuse intrinsic pontine glioma (DPIG) rather than a more common glioma/brain tumor is not clear. As such, the impact of the study is diminished. Lastly, this method does not appear to be high throughput. It seems like a relatively slow process, and thus doesn't add very much to the papers from the 2000s that, while pioneering bioprinting, did so by a very slow spheroid deposition methodology.

We would like to thank Reviewer #1 for their constructive feedback.

With respect to the length of the manuscript, we have made extensive text edits to better contextualize our work. These new text edits are listed in the numbered changes below. We would also like to note that the maximum word count for the main text of a manuscript published by Nature Communications is 5,000 words. The main text of our first submission was approximately 4,500 words. The main text of this revised manuscript is 5,400 words.

With respect to the novelty and utility of our manuscript, we respectfully disagree and believe that the work here is significantly different from all spheroid bioprinting manuscripts to date and addresses a critical unmet need in the field. Briefly, to date, no bioprinting platform has been shown to be able to control the positioning of hiPSC-derived organoids. We apologize that we did not adequately distinguish between spheroids and organoids in our original text. Unlike spheroids, which are amorphous aggregates of cells, organoids have tissue-mimetic structures that self-organize from multiple differentiated cell-types that emerge over time during prolonged *in vitro* culture. As we demonstrate in Figure 1 and Supplementary Figure 4, these organoids undergo severe local deformation in response to the mechanical forces inherent in aspiration and extrusion. In new Supplementary Figure 5, we show that this deformation disrupts the otherwise highly conserved cellular organization within neural organoids.

Taken together, these data underscore the notion that previously reported spheroid bioprinting approaches are not suitable for the bioprinting of hiPSC-derived organoids. To better highlight this crucial difference to future readers, we have changed the STAMP (Spheroid Transfer Assisted by Magnetic Printing) moniker to better reflect the unique capacity for this printing platform to control the spatial positioning of organoids. We now refer to our magnetic bioprinting platform as Spatially Patterned Organoid Transfer (SPOT). We provide further details about the manuscript novelty in response to comment #5 below.

In response to the choice of DIPG as the primary tumor type, we believe that under-studied pediatric diseases, especially those with strikingly poor prognoses such as DIPG, are critically important to study. That being said, the SPOT platform is capable of constructing assembloids from a wide range of organoids. To demonstrate its capacity to bioprint other brain cancers, we have included a new Supplementary Figure 21 wherein we leverage SPOT to construct assembloids comprised of hiPSC-derived neural organoids and three other distinct brain cancers: pediatric glioblastoma, adult glioblastoma, and anaplastic oligodendroglioma. Additionally, in comment #6 below, we provide further rationale for the selection of panobinostat as a therapeutic agent to screen with our neuro-DIPG assembloids.

Finally, we respectfully point out that we do not claim in our manuscript that SPOT is “high-throughput”. The method is scalable and has the potential to be fully automated. We further demonstrate these capabilities through the inclusion of two new Supplementary Videos and two new Supplementary Methods. In its current iteration, SPOT bioprinting has a throughput that is similar to that reported in the recent aspiration-based printing approaches for spheroids (Ayan et al. Science Advances 2020 and Daly et al. Nature Communications 2021). However, neither of those approaches, nor any of the approaches described by the manuscripts from the 2000s (see table in comment #5), would be able to print hiPSC-derived organoids to form neural assembloids. We further discuss scalability, speed, and automation in response to comments #3 and #6 below.

1. There are some typos/grammatical errors throughout; Lack of simple words such as “of”, “the”, etc., that should be double checked.

We have thoroughly read and re-read the manuscript text and corrected any instances of grammatical mistakes.

2. The introduction is extremely short. 3D printing is much broader than simply placing spheroids/organoids. Moreover, there are a plethora of bioactive bioinks that have contributed to the field. None of this is discussed as background. There is also only slight discussion of the need for the presented technology in the field of neurology. Nothing in terms of neuro-oncology (gliomas) is presented. Overall, the introduction does a poor job at setting up the rest of the paper and should thoroughly overhauled.

We agree that the Introduction was concise and have made extensive edits, reproduced below, in an effort to better contextualize our work. In particular, we have taken the reviewer’s recommendation (from comment #5) to highlight pioneering work from the mid 2000s, which introduces the concept of printing spheroids.

We have added the following text in response (page 3):

... in their native environment. **Human neural organoids, three-dimensional (3D) stem cell-derived cultures that self-organize and exhibit tissue-mimetic cytoarchitecture and physiology, have been shown**

to recapitulate facets of brain development *in vitro* (Pasca et al. Nature Methods 2015, Sloan et al. Neuron 2017, Marton et al. Nature Neuroscience 2019, Gordon et al. Nature Neuroscience 2021) and are beginning to reveal mechanistic insights into disease etiologies (Pasca et al. Nature Medicine 2019, Khan et al. Nature Medicine 2020). To model cell-cell interactions and circuit formation in the developing brain, multiple neural organoids have been fused into single integrated tissues known as neural assembloids (Birey et al. Nature 2017, Bagley et al. Nature Methods 2017, Xiang et al. Cell Stem Cell 2017, Miura et al. Nature Biotechnology 2020, Andersen et al. Cell 2020, Kasai et al. Cell Reports 2020, Fligor et al. Stem Cell Reports 2021, Birey et al. Cell Stem Cell 2022). Conventionally, neural organoid fusion ...

... spatial arrangement of spheroids and organoids. Early descriptions of spheroid bioprinting demonstrated the layer-by-layer extrusion of cellular aggregates or cylindrical rods (Boland et al. The Anatomical Record 2003, Jakab et al. PNAS 2004, Jakab et al. Tissue Engineering Part A 2008, Norotte et al. Biomaterials 2009, Skardal et al. Biomaterials 2010). While pioneering, these approaches employed primary cell spheroids that were devoid of internal cytoarchitecture, generally limited in diameter to under 500 μm , and expected to exhibit standardized sizes such that nozzle clogging was obviated (Mironov et al. Biomaterials 2009). Since then, the printing of organ building blocks (OBBs) has been broadly categorized into two distinct ...

Here, we leverage SPOT to control the spatial position of OBBs in two classes of neural assembloids. Firstly, for assembloids employed in studies of neurodevelopmental phenomena, we leverage SPOT to facilitate the construction of assembloids composed of dorsal and ventral forebrain organoids, which mediate *in vitro* studies of the migration and integration of interneurons into the cortex (Birey et al. Nature 2017, Bagley et al. Nature Methods 2017, Xiang et al. Cell Stem Cell 2017, and Birey et al. Cell Stem Cell 2022). Specifically, we demonstrate the potential for SPOT to create complex cellular geometries that would not have been possible with current neural assembloid construction techniques. Secondly, for assembloids employed in translational studies of disease progression and drug efficacy, we leverage SPOT to create tissues in which human brain tumor organoids are integrated into neural organoids. Organoid-based cancer models have emerged as a promising platform for maintaining inter- and intratumoral heterogeneity, enabling *ex vivo* investigation of patient-specific tumor progression (Tuveson and Cleavers Science 2019, LeSavage et al. Nature Materials 2022). To date, two main approaches have been developed for recapitulating the tumor-host cellular microenvironment *in vitro*: (i) by leveraging genetic engineering strategies to induce oncogenic mutations, and (ii) by co-culturing tumor cells with organoid models of the tissue of origin or the tissue of metastasis. While these approaches permit temporal control over the interactions between tumor and host tissue, the SPOT platform enables control of the spatial dynamics of infiltration and, as such, could serve as a complementary approach for building translational *ex vivo* models of cancer.

3. In the Results, the authors lead with a comparison of spheroid diameters between types of spheroids at an arbitrary time point (day 25). Of course as the diameter increases, their mass will, and subsequently the minimum pressure required to lift the spheroids using AAB will increase. This is all quite obvious. Why not limit the timeline for neural organoids and print them while they are smaller in size? There is also one of the original bioprinting methods pioneered by the Forgacs lab that uses spheroids printed from microcapillary tubes that gets around the limitations of AAB. This method is also relatively low throughput, but probably not much more than AAB.

We will address this comment in two parts, first focusing on the timepoints chosen for neural organoid characterization and assembly and then focusing on the throughput of the SPOT platform.

First, with respect to the timing of neural organoid characterization and bioprinting, before day 25, the growth factors and small molecules added throughout the differentiation process are different for dorsal versus ventral forebrain fates (Pasca et al. Nature Methods 2015 and Birey et al. Nature 2017). We therefore selected day 25 as our first time point (at which time the organoids are already 1-2 mm in diameter). Prior to day 46, neural organoids are still undergoing patterning, differentiation, and maturation to acquire their region-specific cell fates. Previously published literature has identified the optimal time window of organoid fusion to be between days 50 and 90 of organoid differentiation (Sloan et al. Nature Protocols 2018 and Miura et al. Nature Protocols 2022). Therefore, to match previously published, well-validated assembloid protocols, we chose timepoints between days 50 and 100 for printing.

We have added the following text to clarify when, with respect to differentiation day, hiPSC-derived neural organoids are ready to be printed (page 5):

... and ventral (subpallium) forebrain organoids. These organoids exhibit canonical markers of dorsal progenitor and ventral forebrain cell fate (Supplementary Fig. 1). Previous studies have observed that the optimal time window during which such organoids should be fused is between days 50 and 90 of differentiation (Sloan et al. Nature Protocols 2018 and Miura et al. Nature Protocols 2022). Compared to MSC spheroids ...

Secondly, with respect to the throughput of the SPOT platform, we have added several demonstrations of the potential of this technology to be automated. In two new Supplementary Videos, we show how G-code can be used to autonomously control the (1) extrusion of the magnetic ink over individual microwells, (2) movement of the magnetized rod between said microwells and the reservoir, and (3) simultaneous switching of the electromagnetic field on and off. Additionally, in two new Supplementary Methods, we provide the G-code scripts for interested readers to repeat the ink extrusion, magnetic rod movement, and switching of the electromagnetic field. We view the SPOT approach as a complementary technology to previous “pick-and-place” bioprinting works including AAB. Unlike AAB or continuous bioprinting approaches, SPOT allows for positioning individual neural organoids into assembloids with high spatial control without compromising their structural integrity. While doing so, the SPOT technology is also able to maintain a similar potential as other “pick-and-place” approaches to be applied to high-throughput assays with automation. The movement of the magnetized print head to transfer four organoids into a four-part assembloid takes approximately 2.5 min. Thus, for our chip designed for four assembloids each composed of four organoids, the assembly could be completed within 10 min. The printer movement required to create these structures would be similar between all “pick-and-place” bioprinting approaches, including AAB. The process of extruding the magnetic ink into each well before the transfer process takes only ~7 seconds/well. To demonstrate these points, we have added the following information to the manuscript:

Supplemental Video 1: Automated magnetic ink extrusion

Supplemental Method 1: G-code for automation of magnetic ink extrusion

Supplemental Video 2: Automated control over magnetic rod movement and electromagnetic field

Supplemental Method 2: G-code for automation of magnetic rod movement and electromagnetic field control

We have also added the following text to both introduce these supplemental materials and better describe the throughput of the SPOT platform (page 8):

... to activate the electromagnet (Supplementary Figure 14c). Here, to support the potential for SPOT to be automated, we demonstrate G-code mediated control of the (i) extrusion of the magnetic ink over individual microwells, (ii) movement of the magnetized rod between said microwells and the reservoir, and (iii) simultaneous switching of the electromagnetic field on and off (Supplementary Vid. 1 and 2). We also provide the accompanying G-code scripts (Supplementary Methods 1 and 2). When taken together ...

4. Figure 2. The requirement for the support scaffold to be directly adjacent to the wells in which the spheroids/organoids form are a massive limitation in the translational nature of this technology.

We thank the reviewer for identifying this potential misunderstanding regarding the design of the custom chip we designed to facilitate SPOT. We agree that the requirement for the support scaffold to be directly adjacent to the wells may pose a limitation to translational applications. However, the support scaffold does not need to be directly adjacent as long as both the organoids and the support bath are submerged within the same body of liquid and the surface tension of the liquid (in this case, cell culture media) is never broken throughout the printing process. If this requirement is met, any configuration of microwell and support scaffold would be amenable to SPOT. Moreover, with respect to the construction of these chips, the ubiquity of plastic 3D printers ensures that it is straightforward to fabricate molds for chips of any desired geometry.

We have included the following text within the Results to clarify this point (page 6):

... into the CNF support scaffold. While the CNF support scaffold is directly adjacent to the organoids in the setup shown here (Fig. 2a, 2b), any configuration in which the organoids can be transferred while remaining submerged within cell culture medium is amenable to SPOT. Importantly, the final position ...

5. Figure 3. This figure is reminiscent of data published by the Forgacs, Mironov, Markwald, and Prestwich labs between 2003 and 2009. It's not clear how this new low throughput method of depositing spheroids has basic science or translational impact.

We thank the reviewer for identifying a series of pioneering papers which first explored the concept of printing spheroids. To address this comment, we have created a table (below) which identifies and summarizes these papers.

Year	Corresponding Author	DOI	Cell Type	Method
2003	Markwald	10.1002/ar.a.10059	Bovine aortal endothelial cells	Aggregates of cells (540 µm diameter) were printed (extruded) layer-by-layer on collagen or thermo-reversible gels
2004	Forgacs	10.1073/pnas.0400164101	Chinese hamster ovary cells	Aggregates of cells (500 µm diameter) were manually placed within a "groove" cut into a disk of polyHPMA or into a partly solidified collagen layer
2008	Forgacs	10.1089/tea.2007.0173	Chinese hamster ovary cells, chicken atrio-	Cell spheroids (300 - 500 µm diameter) were aspirated into

			ventricular cushion tissue buds, human endothelial cells	capillary micropipettes and printed (extruded) layer-by-layer onto collagen bio-paper
2009	Forgacs	10.1016/j.biomaterials.2009.06.034	Chinese hamster ovary cells, human umbilical vein smooth muscle cells, human skin fibroblasts	Cell spheroids (300 - 500 μ m diameter) were formed into cylinders (300 - 500 μ m diameter with a length up to 7 cm) within non-adhesive molds, aspirated into capillary micropipettes, and printed (extruded) layer-by-layer simultaneously with agarose rods
2010	Prestwich	10.1016/j.biomaterials.2010.04.045	Murine fibroblasts	Single cells were encapsulated within a PEGTA hydrogel and drawn into microcapillary tubes to create 500 μ m diameter cylinders and subsequently printed (extruded) layer-by-layer simultaneously with agarose rods
2015	Mironov	10.18063/IJB.2016.01.007	Human dermal fibroblasts	Cell spheroids (up to 200 μ m in diameter) are aspirated into a pipette and printed (extruded) onto electrospun polyurethane with control of spheroid placement in 2D
2017	Mironov	10.1088/1758-5090/aa7fdd	Murine embryonic thyroid explants and allantoic tissue	Cell spheroids (400 - 500 μ m) were loaded into a syringe and moved, through a series of pistons resembling a turnstile, through a bioprinter which printed (extruded) a single spheroid at a time onto a collagen bed in 2D

The SPOT platform was designed to facilitate the spatially controlled construction of assembloids wherein the constituent building blocks are not amenable to conventional approaches for spheroid bioprinting. Compared to the work described above, the SPOT platform exhibits five key differences, which both demonstrate its novelty and mediate its ability to contribute to basic and translational science:

1. Constituent Building Block: Spheroids vs. Organoids - The manuscripts highlighted above and current state-of-the-art approaches (as described in Ayan et al. *Science Advances* 2020 or Daly et al. *Nature Communications* 2021) utilize cell spheroids as the building blocks for their fused tissue constructs. As described in Pasca et al. *Nature* 2022:

“The term spheroids can be used to describe a cellular system obtained by combining in 3D culture one or more separately patterned cell types that have limited self-organization properties. By contrast, in organoids, the cells co-develop through self-organizing and interaction, resulting in cells that differ from mixing separately generated cells.”

Here, we utilize human iPSC-derived neural organoids. These brain region-specific hiPSC-derived organoids are differentiated over the course of 50+ days and progress from pluripotent stem cells to neuroepithelial aggregates to complex structures with tissue-mimetic cytoarchitecture and physiology that resemble various stages of the developing human cortex. Unlike spheroids, which do not exhibit internal cellular arrangements, these organoids contain concentric regions of cells which emulate the early developmental stages of the human cortex (Pasca et al. Nature Methods 2015). Specifically, by day 50 of differentiation, ventricular zone-like structures with PAX6-expressing cells surrounding lumens delimited by NCAD-expressing cells are enriched for PH3-expressing actively dividing progenitors and surrounded by TBR2-expressing intermediate progenitor cells. Over the next 90 days, these ventricular zone-like organizing centers become encircled by deep subcortical projection neurons and, subsequently, superficial cortical neurons in a manner that further reinforces the developmental relevancy of these organoids.

In addition to exhibiting a developmentally relevant cytoarchitecture, these organoids have been shown to undergo functional maturation (electrical activity and synapse formation), to produce glial cells which closely resemble human astrocytes (Sloan et al. Neuron 2017), and to contain oligodendrocytes capable of myelination (Marton et al. Nature Neuroscience 2019). Additionally, long-term maturation of these organoids has been shown, through epigenetic and transcriptomic profiling, to reach postnatal stages between 250-300 days (Gordon et al. Nature Neuroscience 2021).

These organoids are not only relevant for studies of basic developmental biology; several recent manuscripts have demonstrated their potential for translational studies. Of note, these hiPSC-derived neural organoids have been used to perform mechanistic studies on the impact of environmental hits, such as oxygen deprivation (Pasca et al. Nature Medicine 2019), as well as genetic hits, such as the 22q11.2 chromosomal deletion, on corticogenesis (Khan et al. Nature Medicine 2020).

2. Building Block Size - As described in response to comment #3 above, the assembly of these organoids follows the acquisition of specific cell fates that only emerge over protracted culture time (routinely longer than 50 days). As we demonstrate in Figure 1, these hiPSC-derived neural organoids expand in size over time and are significantly larger than MSC spheroids. Across all the manuscripts presented in the table above, the largest size of printed spheroids (or spheroid containing structures) was approximately 500 μm . By day 50, the dorsal neural organoids were approximately 2600 μm in diameter, and the ventral neural organoids were approximately 1500 μm in diameter.

Compared to the manuscripts summarized above, the SPOT platform is able to lift, move, and deposit tissues spanning a significantly wider range of diameters, from relatively small MSC spheroids approximately 300 μm in diameter up to hiPSC-derived neural organoids approaching 3000 μm in diameter. This wider range of workable diameters ensures that SPOT can be utilized across organoid and tissue type.

3. Aspiration and/or Extrusion - The most important distinction between the SPOT platform and conventional spheroid bioprinting approaches (ranging from those described in the table above to more recent demonstrations including Ayan et al. Science Advances 2020 and Daly et al.

Nature Communications 2021) lies in the mechanism used for lifting and depositing the organoids. Unlike previously published approaches, SPOT leverages the magnetic force between the nanoparticles embedded within a hydrogel and an iron rod controlled by a 3D printer. To date, conventional spheroid printing is predicated, in large part, on pressure-mediated aspiration and/or extrusion. These two processes impart a force on the surface of tissues. This localized force may induce localized deformation depending on the magnitude of the force and the apparent surface tension of the tissue.

For spheroids, tissues which do not harbor biologically-relevant cytoarchitectures, a degree of deformation does not substantially affect their integrity as a model. However, for organoids that do exhibit conserved cytoarchitectures, such as neural organoids (Pasca et al. Nature Methods 2015), liver organoids (Guan et al. Nature Communications 2021), and intestinal organoids (Gjorevski et al. Science 2022), such deformation undermines their utility. In Figure 1 and Supplementary Figure 4, we demonstrate that even relatively minimal aspiration force imparts dramatic local deformation to the surface of neural organoids. Moreover, as quantified in Supplementary Figure 4, 20% of neural organoids undergo complete deformation with said force. More importantly, every organoid that was not completely deformed exhibited substantial local deformation. As we demonstrate in new Supplementary Figure 5, such local deformation disrupts the ventricular zone-like structures within neural organoids. By disrupting these structures at early timepoints, this deformation has the potential to preclude the downstream formation of concentric layers of cortical neurons, a key feature of the cytoarchitecture of the developing human brain.

This new supplementary figure is reproduced below along with new text describing the data (page 6):

Supplementary Figure 5. Aspiration disrupts the internal cytoarchitecture of hiPSC-derived neural organoids.

a. Representative IF images of a neural organoid prior to vacuum aspiration. b. Representative IF images of a neural organoid post vacuum aspiration (6 mmHg).

... distorted their spherical shape following release of the aspiration force (Supplementary Fig. 4a, 4c). Importantly, these macroscopic deformations were associated with striking microscopic changes in cellular organization, namely the disintegration of ventricular zone (VZ)-like structures with PAX6-expressing progenitors radially arrayed around lumens lined by NCAD-expressing cells (Supplementary Fig. 5). Over time, intact VZ-like structures give rise to concentric rings of deep subcortical projection neurons and superficial cortical neurons in a manner that resembles the cortical layers of the developing human brain (Pasca 2015). Given the imperative of a conserved cytoarchitecture ...

Taken together, by obviating the need for aspiration or extrusion, SPOT enables the bioprinting of organoids with conserved internal cytoarchitecture. Moreover, the use of magnetic forces for organoid bioprinting has, to our knowledge, never been previously demonstrated.

4. Continuous vs Pick-and-Place Bioprinting - As described in a recent review (Wolf et al. Cell Stem Cell 2022), bioprinting spheroids and organoids can be categorized as either continuous or pick-and-place. In continuous bioprinting approaches, spheroids and organoids are encapsulated within a bioink or support scaffold and extruded. Of the manuscripts described in the table above, any approach wherein a cylinder of spheroids is first constructed can be considered continuous. More recent approaches such as SWIFT (Skylar-Scott et al. Science Advances 2019) or FRESH (Hinton et al. Science Advances 2015) are also continuous. While capable of creating thick, patterned tissues relatively quickly, continuous printing is limited by its inability to address the positioning of individual building blocks as well as the high cost associated with deriving enough spheroids or organoids (especially when they are hiPSC-derived) to populate the bioink or scaffold. Moreover, as described in a review written by several of the scientists listed in this comment, extrusion-based approaches are susceptible to nozzle clogging when the size of spheroids is not standardized (Mironov et al. Biomaterials 2009). The remaining manuscripts described in the table above as well as more recent approaches demonstrated by Ayan et al. Science Advances 2020 and Daly et al. Nature Communications 2021 can be considered pick-and-place. While lower throughput, these approaches are capable of controlling the specific 3D position of each individual spheroid or organoid. However, none of the approaches described in the table above include a quantification of their precision or accuracy. As we demonstrate in Figure 3 and Supplementary Figure 15, the SPOT bioprinting platform is precise. Moreover, as organoids are not extruded through a nozzle, the SPOT platform cannot experience clogging. Finally, previously reported pick-and-place methods have been limited to cell spheroids that are 600 μm or less in diameter. As we demonstrate in Figure 4, the SPOT platform is uniquely capable of printing organoids of various diameters (300 - 3000 μm) in a single experiment.
5. Fused Spheroids vs Assembloids - Assembloids, as defined in Pasca et al. Nature 2022, are:
“Self-organizing cellular systems resulting from the combination of a type of organoids with another type of organoids (for example, dorsal forebrain with ventral forebrain) or with different specialized cell types (for example, cortical organoid with endothelial cells) that result in integration.”

Compared to the fused spheroid structures described in the manuscripts included in the table above, assembloids are, by definition, composed of organoids. As described previously, organoids exhibit a collection of features that make them distinct from spheroids. These distinctions make assembloids uniquely suited for investigations into human development and disease. Several recent manuscripts have begun to explore the potential of assembloids.

During human brain development, interneurons emerge in the ventral forebrain (specifically the subpallium) before migrating and integrating into the dorsal cortex (pallium). The integration of these cells represents a compelling developmental phenomenon that would otherwise be inaccessible as 2D neural cultures have been shown to exhibit altered migratory dynamics (Kawaguchi et al. Nature 2017) and probing human brains *in utero* is clearly unethical. Moreover, this process has been implicated in the etiology of neuropsychiatric disorders including epilepsy

and autism spectrum disorder. Recent work (Birey et al. Nature 2017, Bagley et al. Nature Methods 2017, Xiang et al. Cell Stem Cell 2017, and Birey et al. Cell Stem Cell 2022) demonstrated that human interneuron migration, as well as *in vitro* modeling of diseases related to said migration, can be achieved with neural assembloids. In our manuscript, we derive dorsal and ventral forebrain organoids using protocols first introduced in Birey et al. Nature 2017. We then leverage SPOT to control the spatial positioning of these constituent organoid building blocks to create assembloids with complex geometries that would otherwise not have been possible with current neural assembloid construction techniques.

Neural assembloids have also been generated that recapitulate thalamo-cortical (Xiang et al. Cell Stem Cell 2019), hypothalamic-pituitary (Kasai et al. Cell Reports 2020), and cortico-striatal (Miura et al. Nature Biotechnology 2020) axon projections. hiPSC lines derived from patients with Phelan-McDermid syndrome were differentiated into neural organoids and fused into cortico-striatal assembloids. Interestingly, medium spiny neurons in fused assembloids, but not the isolated organoids, exhibited hyperactivity in line with previous reports from murine and human cortical neurons. Finally, neural assembloids composed of more than three constituent organoid building blocks have also been created. In cortico-motor assembloids, hiPSC-derived cortical organoids are fused to hiPSC-derived spinal organoids that are themselves fused to hiPSC-derived skeletal myoblast spheroids (Andersen et al. Cell 2020). In addition to containing mature synapses and neuromuscular contacts, these assembloids exhibited functional connectivity such that optogenetic stimulation of the cortical organoid induced contraction in the fused skeletal muscle.

Given the novelty of assembloids (the first manuscripts describing neural assembloids were published in 2017), a collection of limitations still remains, which limit their potential for basic and translational science. A major limitation, and the limitation we aimed to address with the development of the SPOT platform, is the lack of control over the position of organoids in three-dimensional space. With SPOT, organoid building blocks can be controllably and reproducibly placed in 3D. We foresee this spatial control as a requisite for the construction of complex, multi-dimensional tissues wherein neural circuits are recreated in a dish. From a translational perspective, SPOT harbors great utility in its capacity to create neural circuits from hiPSC lines derived from patients with genetic variants implied in the etiology of neuropsychiatric disorders. Additionally, as we demonstrate in Figure 4 and its accompanying Supplementary Figures, SPOT can also be used in a translational context to control the placement of tumor organoids. We believe specific combinations of tumor organoids and healthy tissue organoids (e.g. ventral-tumor-dorsal or metastatic tumor-dorsal-originating tumor) allow for heretofore unexplored comparisons of cell-cell signaling in cancer infiltration.

In relation to this comment, we have added the following text to the Introduction (page 3):

... spatial arrangement of spheroids and organoids. Early descriptions of spheroid bioprinting demonstrated the layer-by-layer extrusion of cellular aggregates or cylindrical rods (Boland et al. The Anatomical Record 2003, Jakab et al. PNAS 2004, Jakab et al. Tissue Engineering Part A 2008, Norotte et al. Biomaterials 2009, Skardal et al. Biomaterials 2010). While pioneering, these approaches employed primary cell spheroids that were devoid of internal cytoarchitecture, generally limited in diameter to under 500 μm , and expected to exhibit standardized sizes such that nozzle clogging was obviated (Mironov et

al. Biomaterials 2009). Since then, the printing of organ building blocks (OBBs) has been broadly categorized into two distinct ...

In relation to this comment, we have also added the following text to the Discussion (page 12):

Here, we ... develop a novel organoid bioprinting platform, termed SPOT, wherein individual OBBs can be positioned in 3D space with both a high degree of spatial control and the preservation of internal cytoarchitecture. The positioning of these OBBs is achieved through the use of an MNP-laden, bioinert hydrogel that envelops the tissue of interest and facilitates electromagnet-mediated lifting, transfer, and deposition within a hydrogel support scaffold. Within this matrix, OBBs can undergo fusion to create assembloids. With SPOT, we construct neural assembloids to serve as *in vitro* models both of a neurodevelopmental phenomenon, namely the migration and integration of interneurons into the pallium, and of neural disease progression, namely the infiltration of tumor cells.

This novel magnetic bioprinting approach ... without damaging the constitutive OBB.

Magnetic forces have been previously shown to mediate the formation of patterned 3D tissues from single cells in a now commercialized process known as magnetic levitation (Souza et al. Nature Nanotechnology 2010, Haisler et al. Nature Protocols 2013). While magnetic levitation and SPOT both rely on MNPs, there are several key differences between the platforms. Firstly, with magnetic levitation, individual cells are maneuvered into a desired geometry. SPOT mediates the controlled movement of entire spheroids or organoids and is therefore uniquely suited to applications wherein the cytoarchitecture of an OBB is critical to its fidelity as a model. Secondly, magnetic levitation is predicated on the cellular uptake of a bioinorganic hydrogel containing iron oxide, while SPOT temporarily coats the surface of an OBB with an MNP-laden hydrogel. This transient exposure to MNPs limits the potential for OBBs to undergo any MNP-induced alterations in cellular phenotype. Therefore, when compared to this previous magnetic bioprinting approach, SPOT is particularly well suited for constructing assembloids from organoids with conserved cellular arrangements.

SPOT aims to serve as a complementary ... multiple OBBs in three dimensions.

We engineered the SPOT platform to be accurate, scalable, and readily adoptable, and have identified several technical steps that may be of interest to those intending to incorporate it into their experimental workflows. First, the MNP concentration, coating time, magnetic rod diameter, and magnetic field strength must be optimized for the largest OBB in an experiment. Second, while SPOT can accommodate a range of OBB diameters from 300 μm to 3000 μm , it struggles to accurately deposit OBBs under 300 μm . Further optimization of the attachment of such organoids to the magnetic rod may ameliorate this particular limitation.

... wherein the ratio and positioning of each OBB is controllably varied. We envision that combination of the SPOT platform with spatially-resolved single-cell RNA sequencing, multiplexed time-lapse immunofluorescence, and imaging mass cytometry would have the potential to reveal compelling mechanistic insights into the spatiotemporal dynamics of tumor infiltration. Future studies may adopt the platform for ...

6. Panobinostat is a very specific and best case treatment to use. A more convincing study would screen multiple compounds, including clinically-relevant therapies given that many “mechanistically effective”

drugs fail in brain tumors because they cannot cross the blood brain barrier. Moreover, this would better test the scalability of the platform, which while touted as “scalable” seems relatively low throughput.

As mentioned above, we believe the novelty and utility of the SPOT platform lies in its capacity to construct assembloids with a high degree of spatial control over the placement of organoids. As such, to both address this comment (as well as comments raised by other reviewers) and expand upon the translational relevance of this work, we have included a new Supplementary Figure 21 in which we leverage SPOT to construct assembloids comprised of hiPSC-derived neural organoids and three additional distinct brain cancers:

- Pediatric glioblastoma
- Adult glioblastoma
- Anaplastic oligodendrogloma

This new supplementary figure is reproduced below along with new text describing the data (page 11):

Supplementary Figure 21. A collection of brain tumor organoids fused to a neural organoid.

a. Representative BF images of DIPG, pediatric glioblastoma (pGBM), adult glioblastoma (aGBM), and anaplastic oligodendrogloma (AO3) organoids. b. Representative BF images of mScarlet-expressing dorsal forebrain neural organoids fused to primary brain tumor organoids one day post-fusion.

... additional brain region- or tumor-specific models, such as those for pediatric glioblastoma, adult glioblastoma, or anaplastic oligodendrogloma (Supplementary Fig. 21).

As described above, the scalability of the SPOT platform is directly linked to its capacity to be automated. Current state-of-the-art approaches for aspiration-mediated spheroid printing as well as all approaches for neural assembloid construction are performed manually. Full automation of the platform would remove the need for direct human interaction, and, in so doing, increase its scale. Given the new Supplementary Videos and Methods, we maintain that SPOT has the potential to be fully automated and scalable; that said, in an effort to describe our work as accurately as possible, we refrain from using the phrase “high-

throughput” throughout the text. A note on semantics, we believe that a platform can be “scalable” while still being somewhat slow as long as it is automated.

To help make this point to future readers, we have added the following text (page 8):

... to activate the electromagnet (Supplementary Figure 14c). Here, to support the potential for SPOT to be automated, we demonstrate G-code mediated control of the (i) extrusion of the magnetic ink over individual microwells, (ii) movement of the magnetized rod between said microwells and the reservoir, and (iii) simultaneous switching of the electromagnetic field on and off (Supplementary Vid. 1 and 2). We also provide the accompanying G-code scripts (Supplementary Methods 1 and 2). When taken together ...

We agree with the referee that SPOT is well suited for drug screening applications. However, since drug screening for DIPG has already been reported in the literature (as described in the following paragraph), we instead sought to identify an application of SPOT that represents a biologically unexplored question. Specifically, we study whether a promising drug candidate (panobinostat) has altered efficacy against cancerous cells in the presence of healthy neural tissue.

In a screen of 83 potential agents performed in 2015, panobinostat was shown to both restore H3K27 methylation and normalize oncogene expression (Grasso et al. Nature Medicine 2015). As a promising therapeutic for an intractable disease, panobinostat was moved to a series of clinical trials (NCT02717455, NCT03566199, NCT03632317). Follow-up studies demonstrated that resistance to panobinostat emerges in preclinical models of DIPG (Nagaraja et al. Cancer Cell 2017), suggesting a need for additional agents and/or combinatorial treatments. To address this need, a study in 2019 performed multiple combinatorial screens encompassing 9,195 discrete drug-drug combinations (Lin et al. Science Translational Medicine 2020). While these studies demonstrated the clinical potential of panobinostat, they did not explore the effects of panobinostat (or any other potential therapeutics) on adjacent human neural tissue nor did they explore the effects of cellular crosstalk between DIPG and neural tissue on therapeutic efficacy. While it would be potentially interesting to perform a similar combinatorial screen on neural organoids fused to DIPG tumors, we believe such a screen would be prohibitively resource intensive (both with respect to cells and time). We envision SPOT serving a complementary role to such agnostic screens such that once a promising therapeutic is identified in a high-throughput assay (wherein a single tumor organoid resides in a single well and is treated with one or more therapeutics), SPOT is used to create a collection of assembloids to characterize the effects of cell-cell signaling between cancerous and healthy cells in therapeutic efficacy and safety. Figure 4 and Supplementary Figure 20 exemplify such an approach, as we demonstrate, for the first time, a difference in panobinostat efficacy as a function of the metastatic profile of the DIPG tumor when fused to healthy tissue.

To help make this point to future readers, we include the following text (page 10):

... patient-derived models that have helped identify promising therapeutic agents. However, to date, no experimental models have recapitulated the interactions between DIPG and healthy human neural tissue from distinct brain regions. To demonstrate the potential for SPOT to facilitate studies characterizing the interactions between glioma and human neural tissue *ex vivo*, we created assembloids consisting of hiPSC-derived regionalized neural organoids and patient-derived DIPG organoids with distinct metastatic profiles. In addition to probing the infiltration of DIPG, we were interested in studying whether a leading

drug candidate, panobinostat, might have altered efficacy in the presence or absence of healthy neural tissue from distinct brain regions.

7. Like the Introduction, the Discussion is quite brief. It largely just describes the STAMP approach. There is essentially no discussion or interpretation of the data presented.

Our intention was to include discussion and interpretation of data within the Results section immediately following the initial presentation of said data. In so doing, we intended for our Discussion to primarily focus on the unique advantages of this magnetic bioprinting platform compared firstly to previously demonstrated spheroid printing approaches and secondly to manual approaches for creating neural assembloids (the current state-of-the-art for neural assembloids).

That said, we appreciate the reviewer's recommendation and have amended our Discussion with the following text (page 12):

Here, we ... develop a novel organoid bioprinting platform, termed SPOT, wherein individual OBBs can be positioned in 3D space with both a high degree of spatial control and the preservation of internal cytoarchitecture. The positioning of these OBBs is achieved through the use of an MNP-laden, bioinert hydrogel that envelops the tissue of interest and facilitates electromagnet-mediated lifting, transfer, and deposition within a hydrogel support scaffold. Within this matrix, OBBs can undergo fusion to create assembloids. With SPOT, we construct neural assembloids to serve as *in vitro* models both of a neurodevelopmental phenomenon, namely the migration and integration of interneurons into the pallium, and of neural disease progression, namely the infiltration of tumor cells.

This novel magnetic bioprinting approach ... without damaging the constitutive OBB ...

Magnetic forces have been previously shown to mediate the formation of patterned 3D tissues from single cells in a now commercialized process known as magnetic levitation (Souza et al. Nature Nanotechnology 2010, Haisler et al. Nature Protocols 2013). While magnetic levitation and SPOT both rely on MNPs, there are several key differences between the platforms. Firstly, with magnetic levitation, individual cells are maneuvered into a desired geometry. SPOT mediates the controlled movement of entire spheroids or organoids and is therefore uniquely suited to applications wherein the cytoarchitecture of an OBB is critical to its fidelity as a model. Secondly, magnetic levitation is predicated on the cellular uptake of a bioinorganic hydrogel containing iron oxide, while SPOT temporarily coats the surface of an OBB with an MNP-laden hydrogel. This transient exposure to MNPs limits the potential for OBBs to undergo any MNP-induced alterations in cellular phenotype. Therefore, when compared to this previous magnetic bioprinting approach, SPOT is particularly well suited for constructing assembloids from organoids with conserved cellular arrangements.

SPOT aims to serve as a complementary ... multiple OBBs in three dimensions.

We engineered the SPOT platform to be accurate, scalable, and readily adoptable, and have identified several technical steps that may be of interest to those interested in incorporating it into their experimental workflows. First, the MNP concentration, coating time, magnetic rod diameter, and magnetic field strength must be optimized for the largest OBB in an experiment. Second, while SPOT can accommodate a range of OBB diameters from 300 μm to 3000 μm , it struggles to accurately deposit OBBs under 300 μm . Further optimization of the attachment of such organoids to the magnetic rod may ameliorate this particular limitation.

... wherein the ratio and positioning of each OBB is controllably varied. We envision that combination of the SPOT platform with spatially-resolved single-cell RNA sequencing, multiplexed time-lapse immunofluorescence, and imaging mass cytometry would have the potential to reveal compelling mechanistic insights into the spatiotemporal dynamics of tumor infiltration. Future studies may adopt the platform for ...

8. Why is nanocellulose the biomaterial of choice? It is not a native extracellular matrix-derived biomaterial. Why not use a material that actually would be compatible with human biology? Bioprintable and shear thinning ECM-based/inspired biomaterials/bioinks exist.

We thank the reviewer for this interesting question. Cellulose nanofibers were chosen precisely because they are orthogonal to human biology, as the reviewer stated. As described in the Results of the manuscript, these brain region-specific organoids and assembloids are typically cultured in the absence of exogenous biomaterials (Pasca et al. Nature Methods 2015, Birey et al. Nature 2017, Sloan et al. Neuron 2017, Yoon et al. Nature Methods 2019, Marton et al. Nature Neuroscience 2019, Pasca et al. Nature Medicine 2019, Khan et al. Nature Medicine 2020, Miura et al. Nature Biotechnology 2020, Andersen et al. Cell 2020, Revah et al. Nature 2022). Following assembly within a permissive biomaterial support bath, the assembloids can be readily removed from the cellulose nanofiber gel to mirror common assembloid applications that also lack a biomaterial. Importantly, the applications of assembloids fabricated with SPOT are intended to probe the interactions between the organoid building blocks that comprise the assembloid rather than between an individual organoid and the surrounding matrix.

To help make this point to future readers, we include the following text (pages 6):

... which are embedded within a **bioinert** CNF ink biomaterial that encases the OBB.

Although recent efforts have begun to introduce polymers into the medium of regionalized neural organoids, to date, most studies have cultured organoids in suspension without the addition of exogenous biomaterials.

... we sought to identify a material that was cytocompatible, bioinert to mammalian cells, and amenable to on-demand solubilization to release the encapsulated cellular structure after fabrication.

Reviewer #2 (Remarks to the Author):

In this article a new platform called Spheroid Transfer Assisted by Magnetic Printing (STAMP) that aims to improve the biofabrication of 3D tissues by providing temporal and spatial control of cell-cell interactions. The platform consists of an iron-oxide nanoparticle-laden hydrogel and a magnetized 3D printer that can lift, transport and deposit spheroids and organoids. Cellulose nanofibers are used as the biomaterial for encasing organoids with magnetic nanoparticles and as a support hydrogel for maintaining the spatial positioning of organoids to create precisely arranged assembloids. The STAMP platform was used to create assembloids composed of human pluripotent stem cell derived neural organoids and patient-derived glioma organoids. The article is well written, has solid novel idea and supported by robust experimental design and well performed indicating the potential to construct assembloids. I suggest publication of this article with minor revision.

We thank the reviewer for their feedback and for recognizing the novelty of the platform.

1. It worth to also discuss the potential challenge of this technique, for example of the authors experienced any challenges with floating of OBBs?, attachment of multiple OBB to magnetic tip, attachment of OBB to each other etc. also the author may comment on movement of the magnetic tip in Z axis to pick the OBB and how they could define the Z value for the print head considering that OBBs may not have the exact same height or may not distributed as single layer or not as single OBB at the bottom of the container.

As described in response to comments made by Reviewer 1, we have amended our Discussion to better contextualize the SPOT platform. This comment has encouraged us to also include a brief discussion of some of the limitations still present with SPOT (page 13). The text amendments made in this regard are reproduced below.

That said, we would like to use this space to expand upon some of the specific concerns included in this comment.

Firstly, with respect to floating OBBs, no, we do not observe the neural organoids floating in our custom microwell chips. These brain region-specific organoids are relatively dense tissues as they do not contain lumens. As such, when they are placed in the chip, they settle quickly on their own into the microwells. As we demonstrate in Figure 3, Panel C, we do observe a degree of drift in the Z-dimension post-printing into the cellulose support scaffold. This drift is consistent and dependent on the diameter of the organoid. Importantly, we do not observe significant floating (or other settling) of the organoid within the support scaffold as quantified in Figure 3, Panel D.

With respect to multiple organoids potentially adhering to the magnetic rod, we place a single organoid in a single microwell. This, along with the precision of the 3D printer-controlled rod, ensures that only one nanoparticle-coated organoid is lifted at a time. The strength of our magnetic field, depicted in Supplementary Figure 7, is dependent on proximity. As depicted in Supplementary Figure 14, the height of our microwells is sufficiently large and the wells themselves are sufficiently spaced apart, such that organoids in separate wells do not experience high enough magnetic fields and are not drawn to the magnetic rod.

All movements of the magnetic rod are controlled by a 3D printer. Importantly, this allows us to precisely address its position using G-Code. We have included a new Supplementary Video and

two new Supplementary Methods which depict these movements and provide said G-Code. The reviewer brings up an interesting observation with regard to the potential differences in the diameter of organoids within a series of microwells. As noted above, only a single organoid is included in a single microwell. That said, dorsal and ventral forebrain organoids at day 50 of differentiation have different diameters. Moreover, the neural organoids in general are much larger than the DIPG organoids used within this study. As described in a review on spheroid bioprinting approaches developed in the 2000s, these differences in diameter would pose a significant challenge to aspiration and extrusion based printing (Mironov et al. Biomaterials 2009). However, with SPOT, the coating of spheroids and organoids with magnetic nanoparticle-embedded cellulose nanofibers is not influenced by tissue diameter, ranging from 300 μm to 3000 μm . Moreover, in our experience, the magnetic field generated by the electromagnet and conveyed by the magnetic rod are sufficient to lift a wide range of coated tissues with a fixed rod height. That said, if needed, the G-Code that controls the position of the magnetic rod could be adjusted on a per microwell basis to account for particularly larger differences in tissue diameter. OBBs of a diameter under 300 μm were challenging to accurately deposit, although further optimization of the size of the iron rod or the viscosity of the CNF support scaffold could ameliorate this limitation.

We engineered the SPOT platform to be accurate, scalable, and readily adoptable, and have identified several technical steps that may be of interest to those interested in incorporating it into their experimental workflows. First, the MNP concentration, coating time, magnetic rod diameter, and magnetic field strength must be optimized for the largest OBB in an experiment. Second, while SPOT can accommodate a range of OBB diameters from 300 μm to 3000 μm , it struggles to accurately deposit OBBs under 300 μm . Further optimization of the attachment of such organoids to the magnetic rod may ameliorate this particular limitation.

2. The author mentioned that a potential cytocompatible ink should have a zero-shear viscosity to prevent MNP sedimentation and to be shear thinning, however the provided rheological graph particularly for 0.1% CNF doesn't show the viscous behaviour of the material at very low shear to show the zero shear behaviour of the material.

We agree with the reviewer and have performed new measurements of the viscosity of our magnetic ink using a larger, more sensitive parallel plate geometry (40 mm) to clearly distinguish the zero-shear behavior of the various ink formulations. While the previous plot began at a shear rate of 0.1/s, we have now been able to characterize the viscosity down to a shear rate of 0.005/s. Thus, we have been able to demonstrate the viscous behavior of the materials as the shear rate approaches zero.

The revised Figure 2, Panel C is reproduced below:

c. Representative viscosity measurements of inks with 1 wt% MNP and various CNF wt%.

3. Authors are also encouraged to discuss the byproduct of cellulase activity, e.g. oligosaccharides and its potential effect on the OBB viability and functionality.

We thank the reviewer for this interesting question. Neural cells are enveloped by a mesh of glycans, referred to as the glycocalyx, although to our knowledge, cellulase does not react with its constituents. While we are not aware of cross-reactivity between cellulase and mammalian oligosaccharides, we agree that this is an interesting topic for future investigation.

Here, we demonstrate in Supplementary Figure 11, Panel E that treatment of neural organoids with cellulase across 6 hr and 24 hr does not impart any noticeable difference in neural organoid viability. As shown in Supplementary Figure 12, CNF-embedded organoids can be treated with cellulase for three days to remove any residual cellulose adhered to the surface of the organoid. These organoids exhibited smooth borders, remained viable in culture for weeks, and were able to be fused into assembloids. That said, the cellulose support bath is permissive to neural organoid fusion and, if needed, can be diluted with pure DPBS (without cellulase) such that the resultant assembloid can be retrieved and cultured over longer periods of time in suspension. Finally, if preferred, the fused assembloid can be cultured within the support scaffold for weeks without needing to be removed (either through DPBS-mediated dilution or cellulase-mediated degradation).

We have added text to the revised manuscript (page 8):

... 0.5 wt% cellulase removes over 98 % of the material over 72 hours. **In lieu of cellulase-mediated degradation, organoids can be released from their support scaffold through the gentle dilution of the CNF hydrogel with DPBS. Alternatively, organoids may be cultured within the scaffold for protracted periods of time.**

4. It could benefit from more specific examples of the types of multi-region neural assembloids and tumor-host assembloids that can be created using the STAMP platform

We agree with the reviewer and, as described in response to comments made by Reviewer 1, have included a new Supplementary Figure 21 in which we leverage SPOT to construct assembloids comprised of hiPSC-derived neural organoids and three distinct brain cancers:

- Pediatric glioblastoma
- Adult glioblastoma
- Anaplastic oligodendroglioma

The new figure and accompanying text are reproduced below (page 11):

Supplementary Figure 21. A collection of brain tumor organoids fused to a neural organoid.

a. Representative BF images of DIPG, pediatric glioblastoma (pGBM), adult glioblastoma (aGBM), and anaplastic oligodendroglioma (AO3) organoids. b. Representative BF images of mScarlet-expressing dorsal forebrain neural organoids fused to primary brain tumor organoids one day post-fusion.

... additional brain region- or tumor-specific models, such as those for pediatric glioblastoma, adult glioblastoma, or anaplastic oligodendroglioma (Supplementary Fig. 21).

5. In the discussion section the authors mentioned the advantage of STAMP, here it is worth to also discuss the time of the two processes of AAB and STAMP. In addition, the cost of the process considering the time and challenges of the coating process can be also discussed to better justify the position of such technologies.

We view the SPOT approach as a complementary technology to previous “pick-and-place” bioprinting works including AAB. Unlike AAB or continuous bioprinting approaches, SPOT allows for positioning of individual neural organoids into assembloids with high spatial control without compromising their structural integrity. The SPOT technology is also able to maintain a similar potential as other “pick-and-place” approaches to be scalable upon automation. The movement of the magnetized print head to transfer four organoids into a four-part assembloid takes approximately 2.5 min. Thus, for our chip designed for four assembloids each composed of four organoids, the assembly could be completed within 10 min. The printer movement required to create these structures would be similar between all “pick-and-place” bioprinting approaches, including AAB, as the reviewer specifically mentions. The process of extruding the magnetic ink into each well before the transfer process takes only ~7 seconds/well. To demonstrate these points, we have added two new Supplementary Videos, where we show how G-code can be used to autonomously control the (1) extrusion of the magnetic ink over individual microwells, (2) movement of the magnetized rod between said microwells and the reservoir, and (3) simultaneous

switching of the electromagnetic field on and off. Additionally, in two new Supplementary Methods, we provide the G-code scripts for interested readers to repeat the ink extrusion, magnetic rod movement, and switching of the electromagnetic field. These new supplementary materials are included in the revised manuscript as follows:

Supplemental Video 1: Automated magnetic ink extrusion

Supplemental Method 1: G-code for automation of magnetic ink extrusion

Supplemental Video 2: Automated control over magnetic rod movement and electromagnetic field

Supplemental Method 2: G-code for automation of magnetic rod movement and electromagnetic field control

We have added the following text to both introduce the supplemental materials described above and better describe the throughput and cost of the SPOT platform (page 8):

... to activate the electromagnet (Supplementary Figure 14c). Here, to support the potential for SPOT to be automated, we demonstrate G-code mediated control of the (i) extrusion of the magnetic ink over individual microwells, (ii) movement of the magnetized rod between said microwells and the reservoir, and (iii) simultaneous switching of the electromagnetic field on and off (Supplementary Vid. 1 and 2). We also provide the accompanying G-code scripts (Supplementary Methods 1 and 2). When taken together ...

With respect to the cost of the process, all of the materials and supplies used for SPOT are readily available and economical. The 3D printer (Monoprice MP Select Mini 3D Printer V2) is available for <\$200 and the power supply is <\$100. The cellulose nanofibers and the iron oxide magnetic particles can be prepared at scale in-house using common, affordable reagents. Therefore, this technique does not present an additional economical barrier compared to similar “pick-and-place” techniques like AAB, which also require a 3D printer and a modifying element (power supply and electromagnet for SPOT, vacuum aspiration system for AAB).

Reviewer #3 (Remarks to the Author):

Roth et al. developed a novel platform wherein they can spatially control assembloids using magnetic based bioprinting. The authors then exploited these techniques to develop clinically relevant glioma models. The results emphasize the development of the novel protocol with less attention on fully characterizing the pre-clinical model. Nonetheless, overall this is an interesting and meaningful contribution to organoid/assembloid engineering. Technical advancements with organotypic assembloids have broad implications in cancer as well as other fields such as tissue engineering.

We thank the reviewer for their feedback and for recognizing the impact this platform could have on both the field of tissue engineering and translational cancer studies.

Minor considerations for the authors to consider are as follows:

1. One of the main appeals of the work is having spatial control. In section 2.6, it is not clear how the protocol aided in high throughput dorsal-ventral differentiation even when evaluating supplementary figure 12. Further description of this key element would be useful as well as inclusion of how dorsal versus ventral was verified with staining etc.

We thank the reviewer for their suggestion. The reviewer is correct that one of the main appeals of SPOT is the spatial control imparted by this “pick-and-place” bioprinting approach without compromising structural integrity. With respect to the throughput of the SPOT platform, we have added two new Supplementary Videos and two new Supplementary Methods demonstrating the potential of this technology to be automated to support scalable assembloid fabrication and downstream assays. We have added the following text to both introduce the supplemental materials described above and to better describe the throughput and cost of the SPOT platform (page 8):

... to activate the electromagnet (Supplementary Figure 14c). Here, to support the potential for SPOT to be automated, we demonstrate G-code mediated control of the (i) extrusion of the magnetic ink over individual microwells, (ii) movement of the magnetized rod between said microwells and the reservoir, and (iii) simultaneous switching of the electromagnetic field on and off (Supplementary Vid. 1 and 2). We also provide the accompanying G-code scripts (Supplementary Methods 1 and 2). When taken together

...

With respect to the platform aiding in “dorsal-ventral differentiation”, we would like to clarify that the constituent organoid building blocks are first differentiated (over at least 50 days of differentiation) into their respective brain regions prior to loading them on our chip. Once the desired cell fate has been established, we leverage SPOT to spatially control the 3D positioning of these organoids. Following their precise deposition into a hydrogel support scaffold, the organoids (i.e., dorsal forebrain and ventral forebrain) fuse into assembloids. Whether continued organoid differentiation is influenced by cell-cell signaling following fusion is currently unknown.

With respect to verifying the dorsal versus ventral forebrain fate within our hiPSC-derived neural organoids, we have included a new Supplementary Figure, reproduced below with text, showing immunohistochemical staining to validate the dorsal and ventral identities (page 5).

Supplementary Figure 1. Dorsal and ventral forebrain organoids express region-specific markers.

a. Representative IF images of a dorsal forebrain neural organoid stained for dorsal progenitor cell (PAX6) and ventral (NKX2.1) fate. b. Representative IF images of a ventral forebrain neural organoid stained for PAX6 and NKX2.1.

... and ventral (subpallium) forebrain organoids². These organoids exhibit canonical markers of dorsal progenitor and ventral forebrain cell fate (Supplementary Fig. 1). Previous studies have observed that the optimal time window during which such organoids should be fused is between days 50 and 90 of differentiation (Sloan et al. Nature Protocols 2018 and Miura et al. Nature Protocols 2022). Compared to MSC spheroids ...

2. The authors should add timepoints for the glioma assembloids. For example in Figure 4 and/or methods as well as results, the following questions need to be answered: how long were assembloids in culture before adding the HDAC inhibitor? What is the temporal course of the invasion of DIPG into the neuro organoids? Was the temporal course of invasion different between the various groups (i.e tumor origin site, or dorsal/ventral region)?

We thank the reviewer for calling attention to these important questions. We have added clarifying text in the Results to more specifically describe the timepoints of neuro-DIPG assembloid culture and panobinostat treatment. Additionally, we have added a new Methods section to describe panobinostat treatment, reproduced below:

7.22 Panobinostat treatment of neuro-DIPG assembloids

Neuro-DIPG assembloids fabricated with SPOT fused within 24 hours. They were subsequently released from the CNF support bath using cellulase treatment as described above and cultured for one week in suspension, with media changes performed every 3-4 days. After one week in suspension culture, the media was replaced with fresh media containing 200 nM panobinostat. Neuro-DIPG assembloids were cultured in the presence of 200 nM panobinostat for 72 hours, with no media changes, after which samples were fixed in 4% PFA for immunohistochemistry. Control samples were also given fresh media without panobinostat on day 7 and cultured for 72 hours.

In terms of the temporal course of DIPG invasion into the neural organoids, we agree that this is a very interesting question. In our studies, we observed eGFP-positive projections from the DIPG into the neural

organoid one-week post-fusion. In the present manuscript, we do not observe significant differences in the extent of invasion one-week post-fusion for different cellular groups. Future experiments leveraging time-lapse imaging techniques would yield interesting insights into potential differences in the temporal course of invasion between dorsal vs. ventral or pontine vs. frontal lobe, but we believe that is beyond the scope of the present manuscript.

To clarify the timing of DIPG infiltration for the reader, we include the following text below (page 10) and also edited the figure captions of Supplementary Figure 16 and Supplementary Figure 17:

The substitution to methionine in histone H3 at lysine 27 (H3K27M), a hallmark of diffuse midline pediatric gliomas was predominantly observed within the tumor organoid, and robust infiltration of GFP-expressing DIPG projections was observed at the tissue interface **one week post-fusion**.

... fused to a non-fluorescent ventral forebrain neural organoid **one week post-fusion**.

... fused to a non-fluorescent dorsal forebrain neural organoid **one week post-fusion**.

To expand upon the potential for SPOT to mediate investigations into the temporal dynamics of tumor infiltration, we have added the following text to the Discussion (page 13):

... wherein the ratio and positioning of each OBB is controllably varied. **We envision that combination of the SPOT platform with spatially-resolved single-cell RNA sequencing, multiplexed time-lapse immunofluorescence, and imaging mass cytometry would have the potential to reveal compelling mechanistic insights into the spatiotemporal dynamics of tumor infiltration.** Future studies may adopt the platform for ...

Major considerations for the authors to consider are as follows:

3. DIPG as noted by the authors themselves mainly is in midline structures such as the brainstem. Demonstration of invasion into forebrain by a pontine DIPG does not seem very relevant clinically without a comparison to brainstem. Engineering brainstem neural organoids are likely beyond the scope of this manuscript but can the authors consider utilizing cortical glioma samples such as glioblastoma. The authors do have a DIPG that metastasized to the cortex but this is exceedingly rare and lacks robust clinical relevance. The utilization of an invasive primary glial tumor that infiltrates into the cortex seems better suited for this portion of the paper and can augment the DIPG work.

The reviewer brings up several important points.

Firstly, with respect to the clinical relevance of DIPG metastasis into the cortex, we would like to call attention to a 2014 study from the labs of Hannes Vogel and Michelle Monje titled, "Subventricular spread of diffuse intrinsic pontine glioma" (Caretti et al. Acta Neuropathologica 2014). In this manuscript, the authors first call attention to a pair of studies from the early 2000s that identified DIPG infiltration into the supratentorial (i.e., cerebrum, including the cortex) region and leptomeninges (Gururangan et al. Journal of Neurooncology 2006 and Yoshimura et al. Neurologia medico-chirurgica 2003). The authors then go on to demonstrate widespread DIPG dissemination throughout the brain including extension into the frontal cortex in 25% and supratentorial leptomeninges in 25% of patients. Moreover, in 63% of patients, DIPG infiltrated the subventricular zone (SVZ) and frontal horns of the lateral ventricles. Neural organoids between day 52 and 76 of differentiation (the time period during which we perform most of our fusions)

were previously shown to exhibit a high degree of transcriptomic similarity to the human ventricular zone and SVZ (Pasca et al. Nature Methods 2015). Taken together, we believe there is sufficient evidence to suggest that an in vitro model of DIPG infiltration into hiPSC-derived neural organoids is clinically relevant.

We would like to note that DIPG-XIII-FL, our DIPG line that was derived from the frontal lobe of the donor, was itself specifically obtained from the SVZ, directly adjacent to the cortex.

We would also like to note that these organoids exhibit epigenetic and transcriptomic profiles indicative of early neurodevelopmental stages (Gordon et al. Nature Neuroscience 2021). As such, they are likely better suited to modeling of pediatric tumors.

To help inform the reader of the clinical relevance of our DIPG model, we have added the following text (page 9):

... compared to adult high-grade gliomas. While pontine in origin, DIPG has been shown to infiltrate extensively throughout the brain, from the subventricular zone (SVZ) through the frontal cortex (Gururangan et al. Journal of Neurooncology 2006, Yoshimura et al. Neurologia medico-chirurgica 2003, Caretti et al. Acta Neuropathologica 2014). Standardized protocols have facilitated the use ...

... we created assembloids consisting of hiPSC-derived regionalized neural organoids harboring distinct SVZ-like regions and patient-derived DIPG organoids with distinct metastatic profiles.

Secondly, we agree that expanding our platform to include other primary tumors would provide additional evidence of the translational potential of SPOT. As such, we have included a new Supplementary Figure 21 in which we leverage SPOT to construct assembloids comprised of hiPSC-derived neural organoids and three other distinct brain cancers:

- Pediatric glioblastoma
- Adult glioblastoma
- Anaplastic oligodendroglioma

The new figure and accompanying text are reproduced below (page 11):

Supplementary Figure 21. A collection of brain tumor organoids fused to a neural organoid. a. Representative BF images of DIPG, pediatric glioblastoma (pGBM), adult glioblastoma (aGBM), and anaplastic oligodendroglioma (AO3) organoids. b. Representative BF images of mScarlet-expressing dorsal forebrain neural organoids fused to primary brain tumor organoids one day post-fusion.

... additional brain region- or tumor-specific models, such as those for pediatric glioblastoma, adult glioblastoma, or anaplastic oligodendroglioma (Supplementary Fig. 21).

4. The authors discuss the importance of the assembloids in order to study microenvironmental impacts as well as cell-cell interactions. Figure 15 does show invasion of a DIPG organoid into a dorsal neural organoid. What cell types are interfacing with these invading cells? Were they different in the dorsal versus ventral neural organoids or with the pontine or metastatic DIPG organoids? Further demonstration that the assembloids truly can be used for understanding cell-cell interactions would be insightful with consideration to single cell sequencing, multiplex immunofluorescence, proteomics, or other methods that can help delineate these interactions further.

We agree with the reviewer that additional high-throughput assays such as spatially-resolved single-cell RNA sequencing, multiplexed ion beam imaging (MIBI), and imaging mass cytometry would have the potential to reveal compelling mechanistic insights into how and why DIPG infiltrates into hiPSC-derived neural organoids. That said, these assays are all particularly resource intensive and, we believe, outside the scope of this manuscript. We believe that this manuscript will serve as introduction of the SPOT platform (and, more broadly, the concept of magnet-mediated organoid bioprinting) to the tissue engineering community. Future follow-up studies should be able to leverage the platform to create assembloids amenable to the kinds of exciting assays suggested above.

We have added the following text to the Discussion to highlight the potential of such compelling future experiments (page 13):

... wherein the ratio and positioning of each OBB is controllably varied. We envision that combination of the SPOT platform with spatially-resolved single-cell RNA sequencing, multiplexed time-lapse immunofluorescence, and imaging mass cytometry would have the potential to reveal compelling mechanistic insights into the spatiotemporal dynamics of tumor infiltration. Future studies may adopt the platform for ...

In an effort to answer the reviewer's question regarding which cell types were interacting with the invading DIPG cells, we have included a new Supplementary Figure 18 and text reproduced below (page 10):

Supplementary Figure 18. Neural and glial cell types surrounding DIPG organoid infiltration into neural organoids.

a. Representative IF images of an eGFP-expressing frontal lobe DIPG metastasis infiltrating a non-fluorescent dorsal forebrain neural organoid with staining for glial cells (GFAP). b. Representative IF images of an eGFP-expressing frontal lobe DIPG metastasis infiltrating a non-fluorescent dorsal forebrain neural organoid with staining for neural projections (TUBB3).

... extension of GFP-expressing projections deeper into the neural organoid throughout networks of neuronal and glial cells (Supplementary Fig. 17, 18).

The reviewer's question pertaining to differences in DIPG infiltration as a function of DIPG metastatic profile (originating tumor from the pons versus metastasized tumor from the frontal lobe) and hiPSC-derived neural organoid brain region (dorsal versus ventral) is excellent. We are quite interested in the prospect of leveraging SPOT to create assembloids that would better facilitate studies aimed at answering such questions. As shown in Figure 4 and Supplementary Figure 16, we were able to create three-part assembloids wherein: (i) a single brain region organoid was fused to two tumors with different metastatic profiles or (ii) a single tumor was fused to two distinct brain region organoids. Given the potential for inter-organoid heterogeneity, we believe these kinds of three-part assembloids are uniquely suited to mediating studies in which a single variable (e.g., metastatic profile or brain region) is controllably changed. This is a first step towards addressing these types of important biological questions in future manuscripts.

As we note in response to comment #2 above, in our two-part neuro-DIPG assembloids, we do not observe quantifiable differences in the infiltration of DIPG (originating or metastasized) into either neural organoid type (dorsal or ventral). Instead, the translational phenotype we observed was the difference in panobinostat-induced DIPG cell death as a function of the absence or presence of fusion to hiPSC-derived neural organoids. That said, DIPG infiltration remains of great interest, and future studies would be well served by performing time-lapse imaging or allowing the DIPG to infiltrate over extended culture times (i.e., more than seven days prior to drug treatment). As described earlier in response to this question, incorporating high-throughput spatial profiling into such studies would be potentially quite rewarding, although beyond the scope of the present manuscript.

REVIEWERS' COMMENTS

Reviewer #1 (Remarks to the Author):

The authors provided a very comprehensive Response to Reviewers document, added the requested text, and generated additional data to support their technology platform. I believe they have sufficiently addressed all of the reviewer comments at this point.

Reviewer #2 (Remarks to the Author):

The manuscript has been extensively revised by the authors, and in the opinion of this reviewer, the paper is now acceptable.

Reviewer #3 (Remarks to the Author):

The authors have taken into consideration my concerns as well as the concerns raised by the other reviewers. The additions of further information about assembloids, supplementary videos with spatial information, and most importantly adding the preclinical models with various brain cancers has significantly improved the manuscript. I have no further revisions.